# Lightweight error-tolerant edge detection using memristor-enabled stochastic computing

Lekai Song[1], Pengyu Liu [1], Jingfang Pei[1], Yang Liu[1,2], Songwei Liu[1], Shengbo Wang [3], Leonard W. T. Ng [4], Tawfique Hasan[5], Kong-Pang Pun[1], Shuo Gao [3] & Guohua Hu [1] ✉

The demand for efficient edge computer vision has spurred the development of stochastic computing for image processing. Memristors, by introducing their inherent switching stochasticity into computation, readily enable stochastic image processing. Here, we present a lightweight, error-tolerant edge detection approach based on memristor-enabled stochastic computing. By integrating memristors into compact logic circuits, we realise lightweight stochastic logics for stochastic number encoding and processing with well-regulated probabilities and correlations. This stochastic and probabilistic computational nature allows the stochastic logics to perform edge detection in edge visual scenarios characterised by high-level errors. As a demonstration, we implement a hardware edge detection operator using the stochastic logics, and prove its exceptional performance with 95% less energy consumption while withstanding 50% bit-flips. The results underscore the potential of our stochastic edge detection approach for developing efficient edge visual hardware for autonomous driving, virtual and augmented reality, medical imaging diagnosis, and beyond.

In edge computer vision, extracting the image features to enable efficient user-scene interaction and decision-making has been a challenging topic due to the intensive computational workload and constrained computational resources at the edge. In this context, edge detection is employed as a fundamental pre-processing technique to extract the key visual cues, such as shallow features of colour, contour, and texture, for efficient image understanding and initial decision-making[1,2]. However, the conventional edge detection approaches using matrix multiplication and gradient computation in the binary computing domain can still lead to excessive computational workload and latency against edge hardware integration and deployment[3]. For instance, the deterministic nature of binary computing decides high-precision data representation that can be highly redundant for

computation[4,5]. Taking multiplication as an example (Fig. 1a), the function, despite its simplicity, requires large-scale logic circuits and excessive operations. The scalability of the logic circuits and operations, and the resultant latency, can increase considerably as the computation throughput increases[6]. The binary data representation with position-dependent bit weights also makes edge detection highly susceptible to errors[7]. As an example, even a single bit-flip can corrupt the input and output. Bit-flips, as common soft errors in digital circuits and computing[8,9], can be typically induced by noise and interferences[10]. Though advanced error detection and correction techniques, such as parity bit, cyclic redundancy check, and hash function are now widely adopted to address bit-flips, they inevitably incur excessive hardware and computational cost.

[1]Department of Electronic Engineering, The Chinese University of Hong Kong, Hong Kong SAR, China. [2]Shun Hing Institute of Advanced Engineering, The Chinese University of Hong Kong, Hong Kong SAR, China. [3]School of Instrumentation and Optoelectronic Engineering, Beihang University, Beijing, China. [4]School of Materials Science and Engineering, Nanyang Technological University, Singapore, Singapore. [5]Cambridge Graphene Centre, University of Cambridge, Cambridge, UK. ✉e-mail: ghhu@ee.cuhk.edu.hk

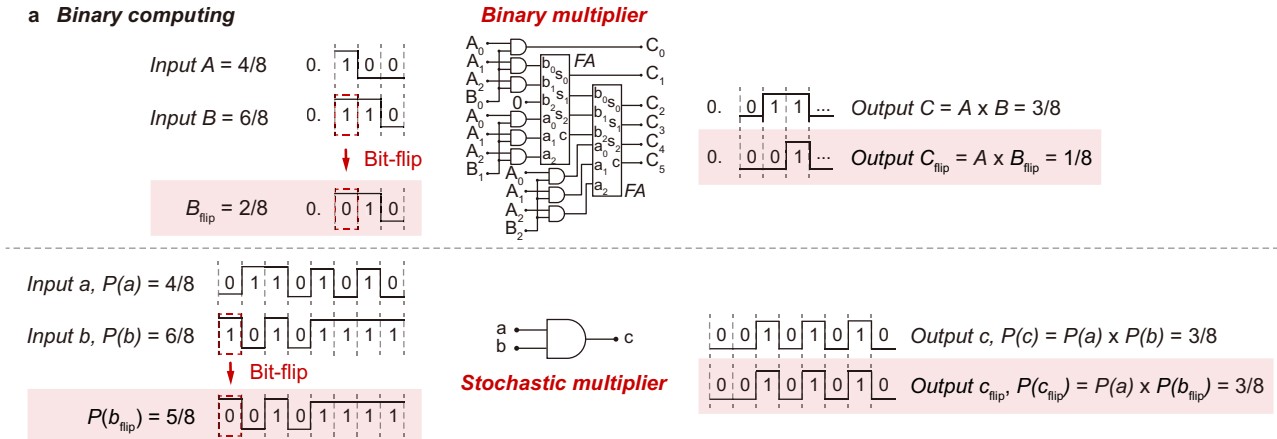

**Fig. 1 | Binary computing vs. stochastic computing. a** Binary computing. Two example 3-bit binary fraction inputs *A* and *B*, representing 4/8 and 6/8, respectively, are computed to yield a binary multiplication output *C* of 3/8. When input *B* undergoes a bit-flip and the value is changed from 6/8 to 2/8, the output is altered from 3/8 to 1/8. *FA* is short for Full Adder. **b** Stochastic computing. Two example 8-bit stochastic number inputs *a* and *b*, with probabilities *P(a)* and *P(b)* of 4/8 and 6/8, respectively, are computed to yield a stochastic multiplication output *P(c)* of 3/8. When input *b* undergoes a bit-flip and the value is changed from 6/8 to 5/8, the output remains at 3/8.

The challenges posed by binary computing put forward a demand for a lightweight, error-tolerant computing paradigm for performing edge detection and other image processing tasks in edge computer vision[11]. Among the various computing strategies, stochastic computing emerges as a promising solution[12]. Unlike binary computing, stochastic computing represents the data as sequences of random 0 s and 1 s bits, known as stochastic numbers, wherein each of the bits holds an equal weight, and the probability of the 1 s bits determines the value of the stochastic numbers[13]. This stochastic and probabilistic nature of data representation allows for the implementation of stochastic computing with lightweight logic circuits and operations. Again, taking multiplication as an example, as illustrated in Fig. 1b, the function can be achieved with one single AND gate, termed stochastic multiplier, and notably, the stochastic multiplier can process stochastic numbers of an arbitrary length without scaling up the logic circuits[14]. Meanwhile, importantly, stochastic computing due to its stochastic and probabilistic computational nature is inherently tolerant of errors. As an example (Fig. 1b), though the occurrence of a random bit-flip corrupts the input, the output can remain invariant. The impact of bit-flips can be even cancelled as the length of the stochastic numbers increases[13]. As such, with the lightweight and error-tolerant computational features, stochastic computing can potentially address the challenges posed by binary computing. Though promising, realising stochastic edge detection (and stochastic computing in general) faces challenges due to the lack of reliable stochastic logics for performing stochastic number encoding and processing[13].

In this work, we present a memristor-enabled stochastic computing approach, and prove its lightweight, error-tolerant stochastic edge detection. We design and realise stochastic number encoders (SNEs) using memristors for stochastic number encoding, and integrate the SNEs with compact logic gates to develop lightweight stochastic logics for stochastic number processing. Harnessing the switching stochasticity of the memristors, the SNEs can encode data into stochastic numbers with well-regulated probabilities and correlations, allowing the stochastic logics to perform bitwise logic operations with statistical probabilities in different correlations. As a practical demonstration of stochastic edge detection, we implement a hardware Roberts cross operator using the stochastic logics and demonstrate its exceptional performance in image contour and texture extractions. Remarkably, the demonstration achieves 95% less energy consumption while withstanding up to 50% bit-flips,

highlighting the lightweight and error-tolerant capability of our stochastic edge detection approach.

## Results
### Stochastic number encoders
SNEs are the units encoding data into stochastic numbers. They have been conventionally realised with electronic circuits (e.g. those based on linear feedback shift registers)[15–17]. However, the circuits are typically on large scales and can lead to considerable computational cost (Supplementary Table 1). As the memristor technology advances, memristors show potential in developing SNEs.

Memristors tend to exhibit stochasticity in switching, originating from the underlying switching mechanisms. For example, due to the stochastic diffusion of the conductive elements, filamentary memristors switch with stochasticity[18]. This characteristic makes memristors promising for realising compact SNEs towards stochastic computing implementation[19–21]. Fig. 2a shows a compact circuit design of SNEs we propose, where each SNE consists of a memristor and a few comparators. By harnessing the switching stochasticity, the SNEs can encode the input data into stochastic numbers – when fed with pulsed inputs $V_{in}$, the memristor is switched stochastically and the output carrying the stochasticity is then binarised by the comparators via the reference $V_{ref}$ for stochastic number encoding with a probability. As such, the probability of the stochastic numbers is well-regulated by $V_{ref}$. The SNEs via convenient circuit reconfiguration can also output stochastic numbers in varying positive and negative correlations, while two or more parallel SNEs can encode uncorrelated stochastic numbers.

To implement the SNEs, we prepare filamentary memristors from solution-processed hexagonal boron nitride (hBN), following our previous report[22]. Briefly, hBN is produced by liquid-phase exfoliation (Supplementary Fig. 1) and used to fabricate memristors in a Pt/Au/hBN/SiOx/Ag configuration (Fig. 2b, c, and Supplementary Fig. 2). This solution-based fabrication approach is scalable with high yield. As demonstrated in Supplementary Fig. 3, the success rate of an array of 12 × 12 memristors is 100% in the sampling test. In typical switching (Fig. 2d), the memristor switches to a low resistive state at the threshold voltage $V_{th}$ as the silver ions diffuse and form conductive filaments, and spontaneously resets to a high resistive state once the bias drops below the hold voltage $V_{hold}$. See also Supplementary Fig. 4 for the ultrafast volatile switching (switching time ~50 ns, and

## a  Stochastic number encoder (SNE)

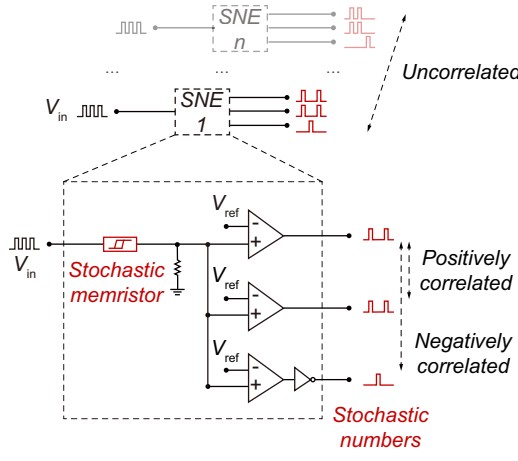

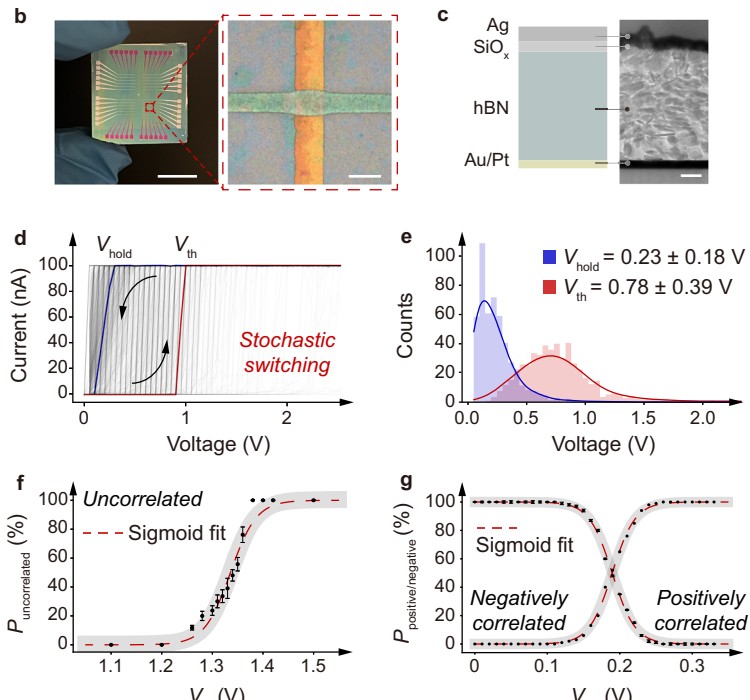

**Fig. 2 | Stochastic number encoder (SNE). a** Schematic SNE, consisting of a memristor and a set of comparators. The output probability and correlation are regulated by both the input $V_{in}$ and reference $V_{ref}$. For negative correlation, a NOT gate is connected to the comparator, and the voltage supply of the NOT gate is synchronised with $V_{in}$ to the memristors to avoid output during the pulse intervals. Independent parallel SNEs are integrated to yield uncorrelated stochastic numbers. See Supplementary Fig. 7 for the hardware realisation of SNE. **b** 12 × 12 memristor array in a crossbar configuration, with a fabrication yield of 100%. A typical device area is ~20 × 20 μm². Scale bar – 1 cm and 20 μm. **c** Schematic and cross-sectional transmission electron microscopic image of a typical memristor. Scale bar – 50 nm.

**d** Current–voltage output from a typical memristor, showing 1000-cycle stochastic yet stable switching with a ratio of ~$10^5$. $V_{hold}$ and $V_{th}$ denote the hold voltage and threshold voltage. **e** Distributions of the measured $V_{hold}$ (0.23 ± 0.18 V) and $V_{th}$ (0.78 ± 0.39 V), along with the corresponding Gaussian fittings. **f** $P_{uncorrelated}$-$V_{in}$ relation of a typical SNE in uncorrelation, fitting sigmoid function $P_{uncorrelated} = 1/(1 + \exp[-38.9(V_{in} - 1.34)])$. The error bar representing the standard deviation at each data point is obtained from 100 repeated samplings, where each sampling consists of 100 consecutive pulsed signal cycles. **g** $P_{positive}$-$V_{in}$ and $P_{negative}$-$V_{in}$ relations of the SNE in positive and negative correlations, fitting sigmoid function $P_{negative} = 1/(1 + \exp[-63.1(V_{in} - 0.19)])$ and $P_{positive} = 1 - P_{negative}$.

relaxation time ~1200 ns) and the ultralow energy consumption (~33 fJ per bit). Due to the stochastic diffusion of the silver ions, the switching exhibits stochasticity in both $V_{th}$ and $V_{hold}$. The volatile switching eliminates the need for any peripheral circuits or excessive resetting for SNE implementation and operation, while the switching stochasticity can be harnessed for stochastic number encoding, leading to compact circuit designs of SNEs.

To assess the stochasticity, we conduct a full sweeping cycling test. The measured current–voltage output exhibits a cycle-to-cycle stochasticity in the switching (Fig. 2d), with $V_{th}$ (0.78 ± 0.39 V) and $V_{hold}$ (0.23 ± 0.18 V) well fitting Gaussian distributions (Fig. 2e). This shows a stabilised cycle-to-cycle stochasticity. We further test the device-to-device stochasticity, and prove a high device-to-device uniformity, with variations of 6.6% in $V_{hold}$ and 7.4% in $V_{th}$ (Supplementary Fig. 3). The uniformity, along with the high fabrication yield, allows for SNE implementation without excess device calibrations or circuit reconfigurations. To evaluate the stochasticity further, we perform the Ornstein-Uhlenbeck process modelling of the measured $V_{th}$ (Supplementary Fig. 5). As demonstrated, $V_{th}$ renders a mean-reverting behaviour with random fluctuations, well-fitting an Ornstein-Uhlenbeck process, i.e. a stochastic process in a dynamical system[19]. This indicates the high-level stability of stochasticity of our memristors in prolonged switching operations, critical for SNE operations. Indeed, the endurance test for over $5 \times 10^6$ cycles proves a highly stable yet stochastic switching of our memristors (Supplementary Fig. 6), outperforming state-of-the-art reports[23–25] and allowing for a

reliable integration of our memristors into circuits for implementing stochastic computing.

We integrate the memristors into the circuits to develop the SNEs (Fig. 2a). When in operation, signals in both digit and analogue forms are first encoded into pulsed inputs, $V_{in}$, and then processed into stochastic numbers via the SNEs, as regulated by $V_{ref}$. See Supplementary Fig. 7 for the hardware realisation of the SNEs. Here we present in Fig. 2f the probability of uncorrelated stochastic number $P_{uncorrelated}$ with respect to $V_{in}$. As $V_{in}$ increases, $P_{uncorrelated}$ is increased, as the memristors tend to be switched on. This proves that the stochastic number occurring at a certain time is probabilistically 0 or 1, and $P_{uncorrelated}$ is determined by $V_{in}$. Particularly, $P_{uncorrelated}$ follows a sigmoidal fitting $P_{uncorrelated} = 1/(1 + \exp[-38.9(V_{in} - 1.34)])$, proving that the SNEs can encode data into stochastic numbers with a well-regulated probability, thereby promising for stochastic computing implementation. In turn, the $P_{uncorrelated}$-$V_{in}$ relation can be employed as a guidance to practically determine $P_{uncorrelated}$ with $V_{in}$. Similarly, we show in Fig. 2g the probabilities of positively and negatively correlated stochastic numbers $P_{positive}$ and $P_{negative}$ with respect to $V_{ref}$. $P_{positive}$ ($P_{negative}$) decreases (increases) as $V_{ref}$ increases in positive (negative) correlation, as $V_{ref}$ serves as the threshold for binarization. Again, $P_{negative}$ follows a sigmoidal fitting $P_{negative} = 1/(1 + \exp[-63.1(V_{ref} - 0.19)])$, and $P_{positive} = 1 - P_{negative}$. See Supplementary Fig. 8 for an example of positively correlated stochastic number encoding. Therefore, the memristor-enabled SNEs prove data encoding into stochastic numbers with regulated probabilities and correlations, facilitating subsequent stochastic logic

development. Here we note the encoding frequency of $V_{in}$ is typically configured as 100 kHz, far below the switching of the memristors (up to 50 ns, or equivalently 20 MHz) and the clock frequency of the digital circuits (~GHz). This ensures that the SNEs can be applied in the implementation of stochastic computing hardware and applications.

## Stochastic logics

We integrate the SNEs with compact logic gates to build lightweight stochastic logics in different correlations. Using stochastic AND logic in uncorrelation as an example, we connect two parallel SNEs to a typical AND gate (Fig. 3a). In this design, the uncorrelated stochastic outputs encoded by the SNEs serve as the inputs to the AND gate, enabling stochastic multiplication of the stochastic outputs. When in operation, based on the demonstrated $P_{uncorrelated}$-$V_{in}$ relation in Fig. 2f, the SNEs are fed with pulsed signal cycles of the corresponding $V_{in}$ to encode uncorrelated stochastic numbers, denoted as $a$ and $b$, with probabilities of $P(a)$ and $P(b)$, respectively. Then, $a$ and $b$ are bit-by-bit fed into the AND gate, yielding a stochastic number output, denoted as $c$, with a probability of $P(c)$. We show in Fig. 3a the corresponding stochastic numbers and probabilities from the experimental hardware test. The statistical relation between the probabilities, i.e., $P(a)P(b) \approx P(c)$, proves that the stochastic AND logic functions as a stochastic multiplier for one-step multiplication of stochastic numbers. Importantly, compared to the binary multiplier in Fig. 1a, this stochastic multiplier significantly simplifies circuit design and reduces the computational cost. Besides, the SNEs can be configured to exhibit positive (negative) correlation, enabling positively (negatively) correlated stochastic AND logic operations (Fig. 3a). The output probability $P(c)$ in the correlated cases is determined by the minimum (maximum) value of $P(a)$ and $P(b)$ instead. Similarly, we build stochastic OR logic in

all three correlations, and it performs different logic operations as designed (Fig. 3b).

Edge detection involves matrix multiplication and gradient computation that normally require large-scale logic circuits and considerable computational operations[3]. In contrast, it is possible to perform absolute-valued subtraction for gradient computation with minimal computational cost using the stochastic logics. Here we propose in Fig. 3c the design of a stochastic XOR logic, consisting of only an SNE and an XOR gate, to perform the function. Specifically, the SNE is fed with pulsed signal cycles of the corresponding $V_{in}$ according to the $P_{positive}$-$V_{in}$ relation in Fig. 2g to encode positively correlated stochastic numbers, denoted as $a$ and $b$, with respective probabilities of $P(a)$ and $P(b)$. Then, $a$ and $b$ serve as the inputs to the XOR gate, and the resultant $P(c)$ satisfies $P(c) \approx |P(a) - P(b)|$. In this case, positively correlated stochastic numbers mean a maximum overlap of 0 s and 1 s, such that the probability for two 1 s or two 0 s is $\min(P(a), P(b))$ or $\min(1 - P(a), 1 - P(b))$. Assuming $P(a) > P(b)$, the stochastic XOR logic outputs $P(c) = 1 - P(b) - (1 - P(a)) = P(a) - P(b)$, and vice versa. This proves the capability of the stochastic XOR logic to perform the absolute-valued subtraction function in only one step. Besides gradient computation, denoising, smoothing, and down-sampling are also essential matrix operations in edge detection. A general approach in performing these functions is to use mean convolutional filters to process the pixels. Here we propose in Fig. 3d (see also Supplementary Fig. 9) the design of stochastic MUX logic to realise a mean convolutional filter.

We present in Supplementary Fig. 10 the pairwise correlations between the inputs of the above stochastic logics in the uncorrelated, positively correlated, and negatively correlated conditions, and summarise the statistical relations between $P(a)$, $P(b)$ and $P(c)$ in Supplementary Table 2. The pairwise correlations and the statistical relations

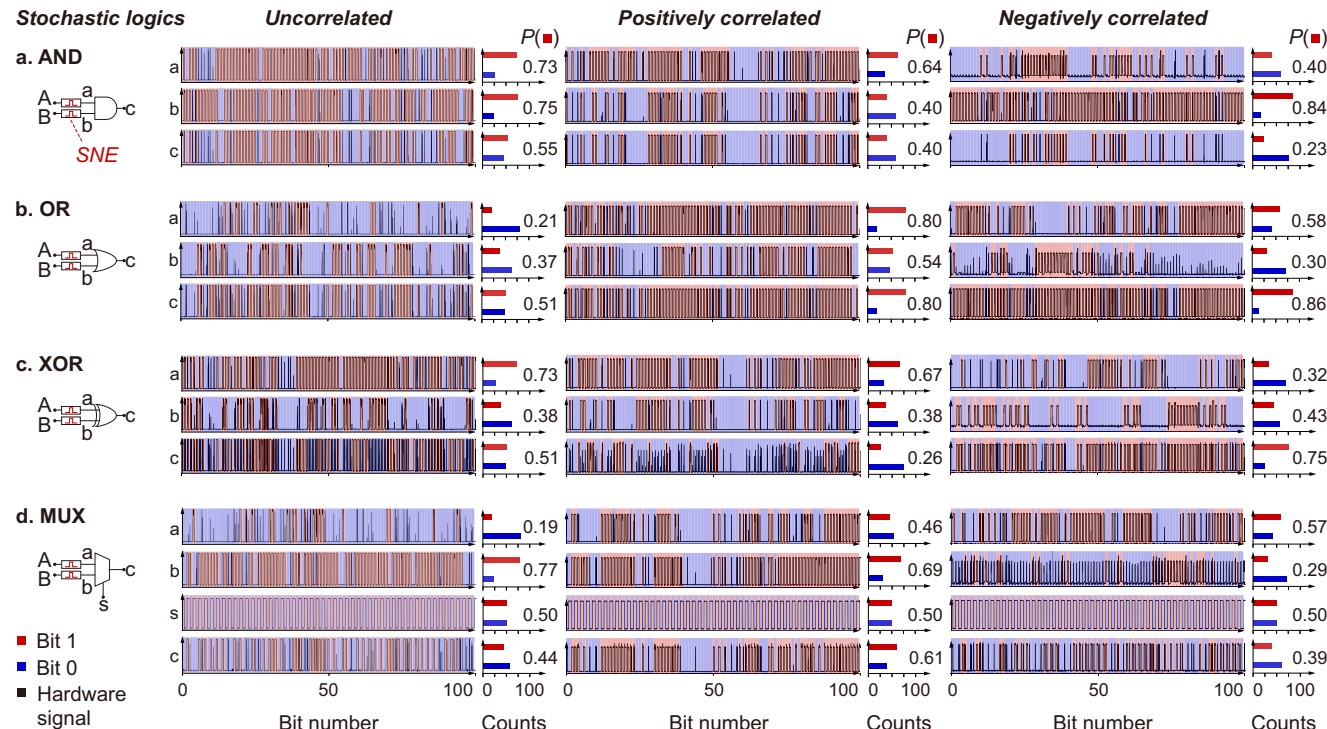

**Fig. 3 | Stochastic logics.** Schematic stochastic logics in uncorrelation implemented with two independent SNEs and **a** AND, **b** OR, **c** XOR, and **d** MUX, and the corresponding circuit tests of the stochastic logic operations. The stochastic logics can be reconfigured in the positive and negative correlations to yield the stochastic logic operations as respectively demonstrated. For stochastic MUX, the frequency of the select $s$ is half of that of the inputs to ensure that both the inputs participate in the logic operations. $P$(red square) represents the probability of the 1 s in the sequences, i.e. the value of the stochastic numbers. The outputs of stochastic logics in uncorrelation, positive correlation, and negative correlation are consistent with the statistical formulas in Supplementary Table 2.

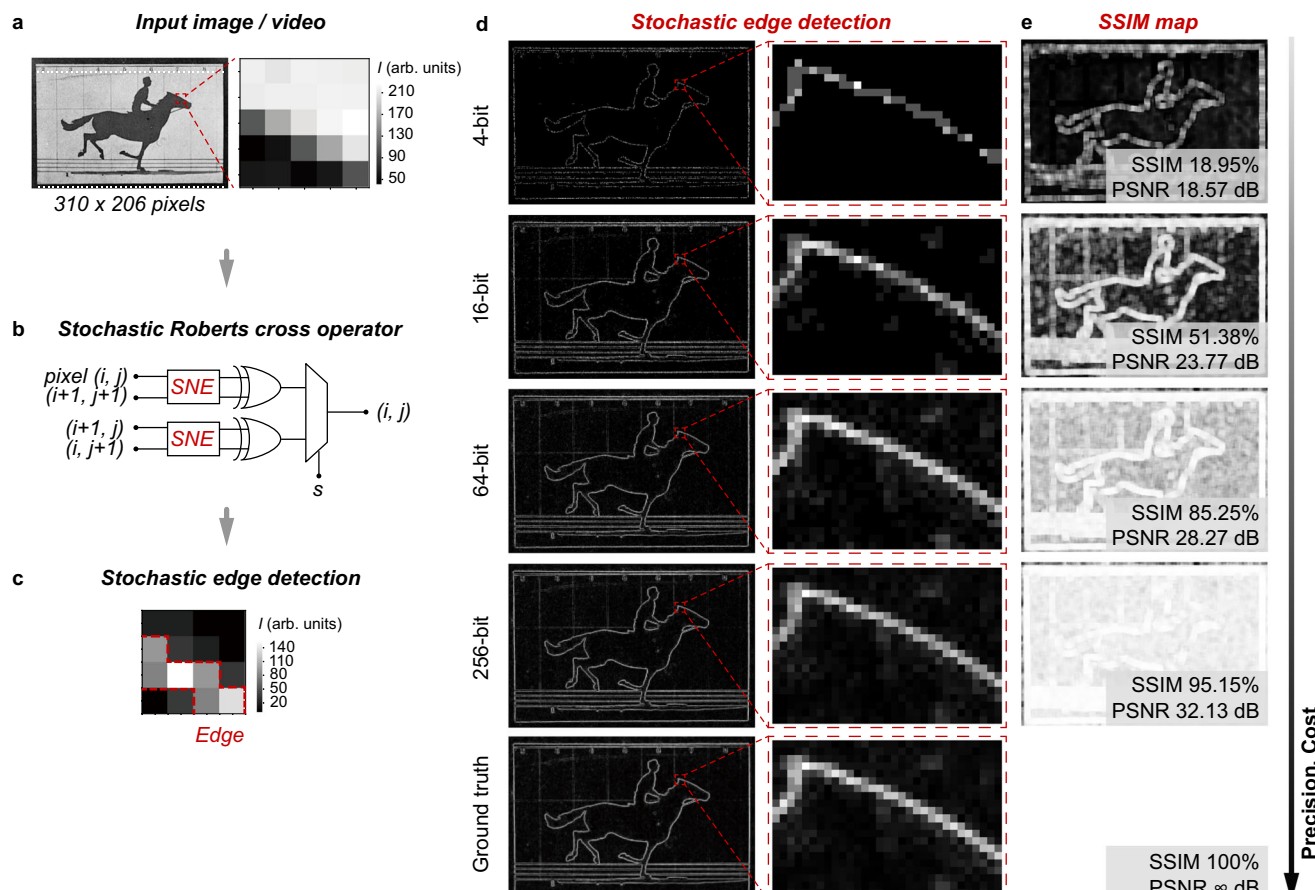

**Fig. 4 | Stochastic edge detection. a** The example image, i.e. the first frame of the The Horse in Motion, for edge detection demonstration. The region as marked is used to illustrate the edge detection process with the operator. The pixels in 0-255 grayscale are encoded into 100-bits. **b** Schematic stochastic Roberts cross operator, consisting of two SNEs, two XORs, and one MUX. See Supplementary Fig. 11 for the hardware realisation of the operator. **c** Gradient map yielded from scanning with the operator, showing successful edge detection. **d** Edge detection of the first frame with the operator, and **e** the corresponding structural similarity index measure (SSIM) maps and peak signal-to-noise ratios (PSNR). The pixels are encoded into 4, 16, 64, and 256-bits as the inputs. The SSIM and PSNR show that the operator using more bits gives higher edge detection precision. For comparison, the edge detection performed using the standard algorithmic method is presented as the ground truth.

confirm that our stochastic logics can work in the desired correlation conditions and conduct the corresponding logic operations for performing edge detection tasks. Pearson correlation is adopted here to quantify the correlations. Note that in the above demonstrations, the stochastic numbers are encoded in 100-bit for illustrative purposes. The bit length can be adjusted to accommodate the different computational precision requirements, given the trade-off between the computational cost and precision.

**Stochastic edge detection**

As discussed, edge detection in the conventional binary computing approaches relies on the use of large-scale logic filters, such as Roberts cross and Sobel operators, leading to significant hardware and computational cost as well as latency[3]. In this context, we propose a hardware stochastic Roberts cross operator using the stochastic logics to address the challenges. Briefly, two SNEs, two XOR gates, and one MUX are integrated to build the stochastic Roberts cross operator. See Fig. 4 and Supplementary Fig. 11 for the design and hardware realisation of the operator.

We apply the stochastic Roberts cross operator in image processing to demonstrate the feasibility of stochastic edge detection. The image for illustrative purposes is captured from the artwork *The Horse in Motion* (Fig. 4a). Here each pixel in 0–255 grayscale is encoded in 100-bits. As shown in Fig. 4b, the stochastic Roberts cross operator is

used to scan over the pixel map to yield a gradient map $(i,j)$ reconstructed from the output stochastic numbers. Specifically, one SNE and one XOR gate work consecutively to yield the $x$ component of the output gradient $(i,j)$, denoted as $|Gx|$, while the other SNE and XOR gate yield the $y$ component, denoted as $|Gy|$. The gradient $G(i,j)$ is obtained by averaging $|Gx|$ and $|Gy|$ using the MUX logic, i.e. $G(i,j) = 0.5(|Gx| + |Gy|)$. The coefficient 0.5 scales the gradient within the original grayscale. As such, as demonstrated in Fig. 4c, scanning with stochastic Roberts cross operator over the marked image region of $5 \times 5$ pixels in Fig. 4a yields a $4 \times 4$ pixeled gradient map that evidently demonstrates successful edge detection, as outlined by the red dashed lines. This confirms the feasibility of the stochastic Roberts cross operator in performing edge detection.

As discussed, the bit length of the stochastic numbers can govern the computational precision. To investigate the impact of the bit length on the stochastic Roberts cross operator for edge detection, we encode the pixels of the image frame in Fig. 4a in 4, 16, 64, and 256-bits, respectively. The edge detection results (Fig. 4d) prove that the edges are successfully detected and recognised in all cases. However, as observed, a longer bit length yields better edge detection. To quantitatively evaluate the performance, we compare the edge detection results with those obtained from the standard algorithmic method. We consider the algorithmic result as the ground truth, and assess the fidelity of the stochastic edge detection using two metrics: the

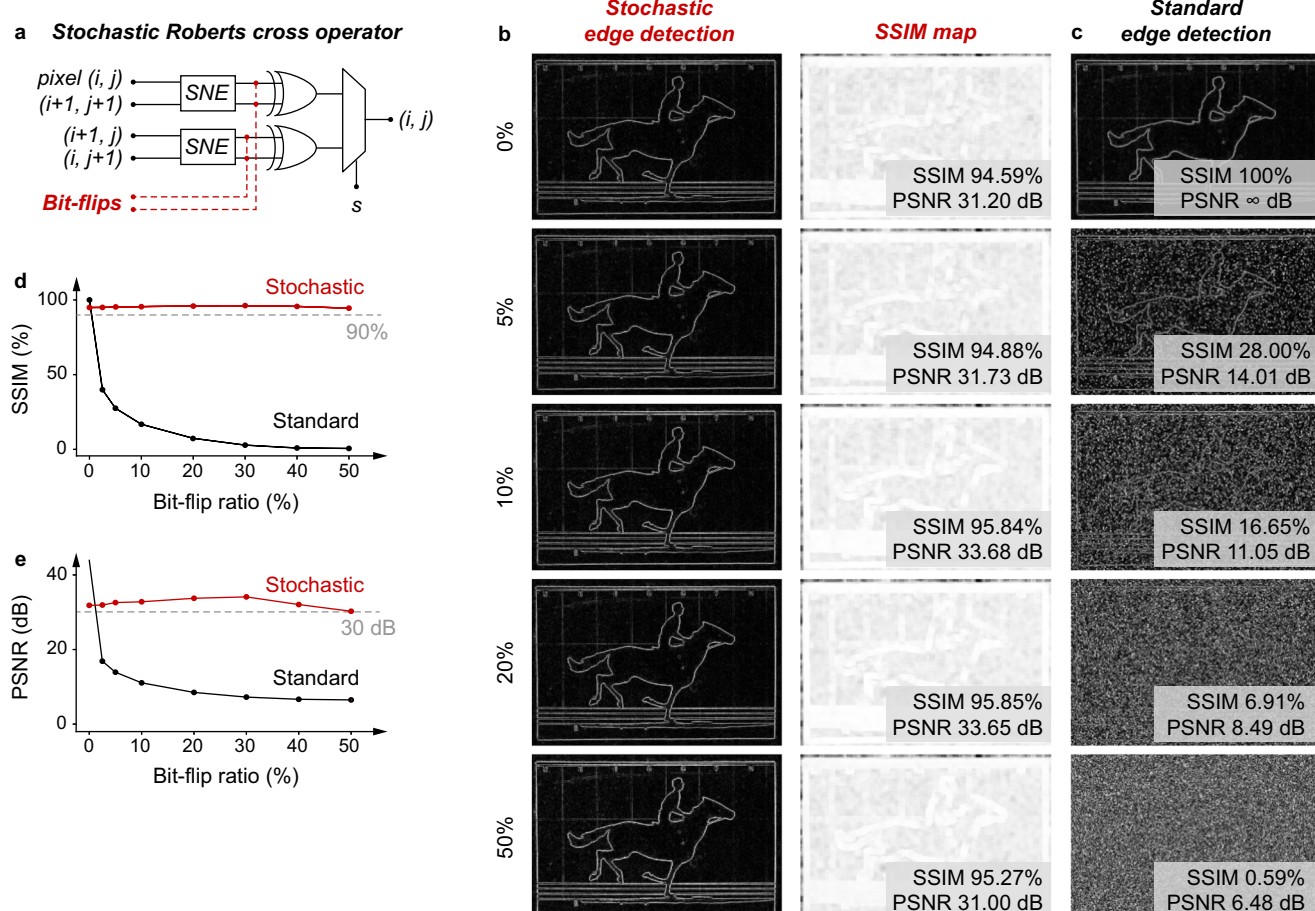

**Fig. 5 | Error-tolerance test. a** Error-tolerance test of the stochastic cross operator, showing bit-flips are injected into the original stochastic numbers for the tests. **b** Stochastic and (**c**) standard edge detection results and the corresponding structural similarity index measure (SSIM) maps of the first frame with bit-flip injection at a ratio of 0%, 5%, 10%, 20%, and 50%. For the stochastic edge detection, the high SSIM (>90%) and peak signal-to-noise ratios (PSNR) (>30 dB) prove that the bit-flip injection does not degrade the edge detection performance. In contrast, a low level of bit-flip injection significantly degrades the performance of the standard algorithmic edge detection. See Supplementary Fig. 12 for the SSIM map of the standard edge detection, and the error-tolerance results at more bit-flip injection levels. Performance comparison in **d** SSIM and **e** PSNR between the stochastic and standard edge detection results.

structural similarity index measure (SSIM) and peak signal-to-noise ratio (PSNR). Here we visualise the loss in performance by the SSIM maps (Fig. 4e), where a brighter pixel indicates a higher similarity to the ground truth, i.e. a better edge detection performance. This thus reveals and confirms that a longer bit length indeed leads to an improved edge detection performance. For instance, the 256-bit achieves a near-ideal performance, with SSIM > 0.95 and PSNR > 30 dB. In contrast, the 4-bit exhibits relatively poor performance, as the limited precision in the 4-bit length fails to accurately encode the 0–255 grayscale. However, as evident in Fig. 4d, e, the 4-bit still successfully detects the edges.

We further investigate the error-tolerance capacity of the stochastic cross operator against bit-flips. Specifically, as illustrated in Fig. 5a, bit-flips from 5% to 50% are injected into the stochastic numbers (in 256-bit encoding). Again, we adopt the SSIM and PSNR metrics to evaluate the edge detection performance. As evident in Fig. 5b, the stochastic Roberts cross operator demonstrates successful edge detection in all levels of bit-flip injections. Notably, the operator even retains an SSIM of >0.95 and a PSNR of >30 dB at a 50% bit-flip injection. In comparison, the performance from the standard algorithmic method substantially degrades at a bit-flip injection of only 5%, with the edges hardly recognised and the SSIM and PSNR significantly decreased (Fig. 5c, d, e). See Supplementary Fig. 12 for the SSIM maps

from the standard algorithmic method, and the error-tolerance results at more bit-flip injection levels. The superior error-tolerance capacity of the stochastic Roberts cross operator originates from the fact that each bit in the stochastic numbers carries an equal weight, and thus the impact of pairs of bit-flips can be cancelled.

### Hardware and computational cost

Exploiting the volatile switching and stochasticity of our hBN memristors, circuits to implement stochastic computing are highly compact. For instance, the SNEs require down to three electrical components to realise, outperforming those based on the conventional electronic circuits and the other memristors (Supplementary Table 1). The compact circuits can not only lead to a much lower hardware cost but also a much less computational cost. To discuss this further, here we compare the energy consumption of our stochastic computing approach with the counterpart in the binary computing domain. Note that the comparison is conducted under the same computational precision, i.e. each $n$-bit binary number is represented by $2^n$-bit stochastic numbers.

For our stochastic edge detection operators, the energy consumption is mainly contributed to the memristors and the remaining comparators and logic gates. In terms of the memristors, the switching energy is estimated as ~33 fJ per bit (Supplementary Fig. 4). As such,

**a**  *Stochastic Roberts operator*

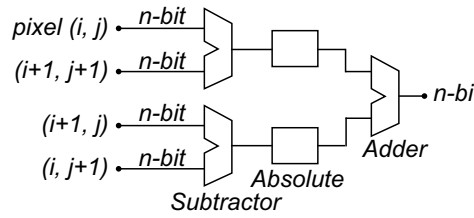

**c**  *n-bit subtractor or adder*

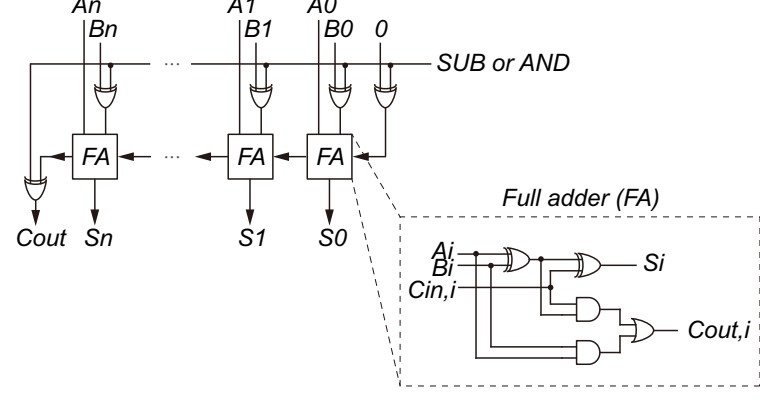

**b**  *Conventional Roberts operator*

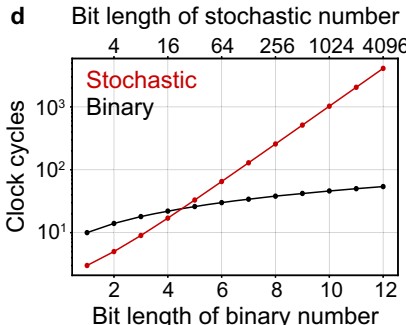

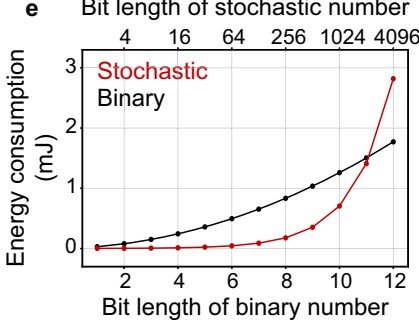

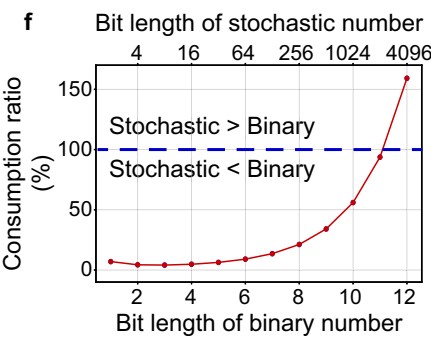

**Fig. 6 | Performance and energy consumption comparison between the stochastic and conventional Roberts cross operators.** Circuit designs of **a** our stochastic and **b** the conventional Roberts cross operator. The $2^n$-bit inputs to our stochastic Roberts operator are stochastic numbers. **c** Circuit design of a $n$-bit subtractor and adder in (**b**). Relation of required **d** clock cycles and **e** energy consumption with respect to the bit length of the binary numbers and stochastic numbers. **f** Energy consumption ratio in (**e**). When the length of the stochastic numbers is within 2048 bits, the energy consumption of the stochastic Roberts cross operator is lower. All comparisons are conducted at an input encoding frequency of 100 kHz.

encoding a $2^n$-bit stochastic number consumes ~$33 \times 2^n$ fJ. Specifically, this estimation assumes the worst-case scenario, where a sufficiently large $V_{in}$ of 2 V is adopted. In fact, a $V_{in}$ of 1.1–1.5 V is adequate to perform the stochastic number encoding (Fig. 2f). In terms of the remaining circuits, here we estimate the energy consumption based on the required counts of logic gates and clock cycles for the stochastic edge detection operators, following $W = k \times T_c \times P$, where $T_c$ is the clock cycle, $k$ is the required counts of $T_c$, and $P$ is the total power of the remaining electrical components, including the comparators and logic gates. For the stochastic Roberts cross operator (Fig. 6a), $2^n$-bit stochastic numbers are processed serially. Therefore, it requires $(2^n + 1)T_c$ and thus, $(2^n + 1)T_c P_{stochastic}$ energy. Given the input encoding frequency of 100 kHz, $T_c$ is 10 µs. To calculate $P_{stochastic}$, we refer to the product power datasheet of the logic gate chips used in our work (Supplementary Table 3).

The energy consumption to perform the edge detection in the binary computing domain is incurred by the conventional edge detection operators. Here we estimate the energy consumption of the conventional Roberts cross operator (Fig. 6b). As illustrated, the operator consists of two $n$-bit subtractors and one $n$-bit adder. Each $n$-bit subtractor and adder can be built using $n$ full adders (FA) and several XOR gates (Fig. 6c). Considering parallel computation, each $n$-bit subtractor and adder requires $(2n + 3)T_c$. Therefore, assuming two subtractors run in parallel, the conventional Roberts cross operator requires $2(2n + 3)T_c$ and thus, $2(2n + 3)T_c P_{conventional}$ energy. Similarly, we assume $T_c = 10$ µs and refer to Supplementary Table 3 to calculate $P_{conventional}$. Here we note that binary computing does not encode the $n$-bit binary number input into stochastic numbers, and additional

circuits and computational operations in practical computing applications are often necessitated to deal with errors.

We present in Fig. 6d–f the comparison between our stochastic and the conventional Roberts cross operators for edge detection. As shown in Fig. 6d, the stochastic operator requires fewer clock cycles to complete edge detection with the stochastic number inputs <16-bits. However, as the bit length increases beyond 16-bits, the stochastic operator requires exponentially increased clock cycles, surpassing the conventional counterpart. Nevertheless, as presented in Fig. 6e, the stochastic operator can maintain a lower energy consumption within 2048-bits. To provide a more intuitive representation of the energy efficiency, we plot the energy consumption ratio of the stochastic operator to the conventional counterpart in Fig. 6f. The results show that with 4-bit stochastic number inputs, the stochastic operator can consume ~95% less energy. Similarly, with 64-bit stochastic number inputs, it can still consume ~90% less energy.

## Discussion
In this work, we have presented a lightweight, error-tolerant stochastic edge detection approach using memristor-enabled stochastic computing. The stochastic computing, realised by harnessing the inherent switching stochasticity in memristors, facilitates the design and implementation of lightweight stochastic logic circuits for performing stochastic edge detection. Particularly, owing to the stochastic and probabilistic computational nature, the stochastic edge detection is well-suited to edge visual scenarios characterised by high-level errors. As a practical demonstration, we show that a hardware stochastic Roberts cross operator can achieve excellent edge detection

with error-tolerance capacity and low hardware and computational cost.

Given the remarkable edge detection performance, the scalability of the memristors, and the compactness of the circuit designs, our stochastic edge detection approach can be readily scaled-up and generalised towards the development of lightweight, error-tolerant edge visual hardware. Particularly, arising from the ultrafast switching characteristic of the memristors, a large-scale stochastic edge detection with 100-bit stochastic number encoding can in principle easily process frames at over 1000 fps, fulfilling the requirements of applications ranging from autonomous driving and virtual/augmented reality to industrial automation and medical imaging diagnosis. To explore the feasibility, we show via simulation large-scale stochastic edge detection of video flows (Supplementary Movies 1–3). Though promising, large-scale stochastic edge detection requires efforts for realising large-scale design, fabrication and integration of the memristors and peripheral electronic circuits, and parallel operation of the large-scale circuits. Amongst this, the success rate and uniformity of the memristors are still key concerns in large-scale manufacturing in current technological advances. A device-to-device non-uniformity can significantly impact the overall operation and performance of the circuits. A system-level analysis of memristors, SNEs, and peripheral circuits may therefore be adopted, e.g. the Process-Voltage-Temperature analysis. Hardware and algorithm codesigns are also needed to address or accommodate the non-idealities, e.g. noises and delays from the memristors and electronic circuits.

## Methods

### Stochastic memristors

Pristine hBN powder and all chemicals are purchased from Sigma-Aldrich and used as received. hBN powder ($10\,g\,L^{-1}$) and polyvinylpyrrolidone ($1\,g\,L^{-1}$) are mixed into isopropanol in a sonication tube in ambient condition. The mixture undergoes 48-h bath sonication at ~10 °C to facilitate exfoliation and dispersion. The dispersion of the as-exfoliated hBN nanoflakes in isopropanol is then centrifuged at 4000 rpm for 30 min to remove insufficiently exfoliated aggregates. The supernatant is carefully decanted and collected. Controlled volumes of isopropanol and 2-butanol are added to formulate a stable hBN ink in isopropanol/2-butanol (90/10 vol%), with a concentration of ~$1\,g\,L^{-1}$. In a typical process, the memristor is fabricated in a vertical Pt/Au/hBN/SiO$_x$/Ag configuration, where hBN is deposited by slot-die coating, the SiO$_x$ layer (10 nm) is deposited by electron beam evaporation, and the metal electrodes (5/15 nm Pt/Au and 30 nm Ag) are patterned by photolithography and deposited by electron beam evaporation. 10 nm SiO$_x$ minimises wash-off of the deposited hBN during the photolithographic patterning process and thereby increases the yield. The device substrate is Si/SiO$_2$. The slot-die coater is Ossila L2005A1. The evaporator is IVS EB-600. During device fabrication, the hBN layer after deposition is baked at 200 °C for 2 h. For a typical memristor, given the thickness difference between the SiO$_x$ (10 nm) and hBN (~260 nm) layers, the switching behaviour is governed by the formation and rupture of silver filaments in the hBN layer.

### SNEs and stochastic logics

To build the SNEs and stochastic logics, the memristors are tested on a probe station and connected to the logic gates and other electronic devices on a breadboard. Tektronix Keithley 4200A-SCS parameter analyser with pulse measure units is used to measure the electrical characteristics of the memristor. Siglent arbitrary waveform generators and digital storage oscilloscope are used to output the signals and measure the output waveforms. To endow the stochastic numbers with a certain probability, based on the demonstrated $P_{uncorrelated/positive/negative}$-$V_{in}$ relation in Fig. 2f, g, each SNE are fed with $n$ pulsed signal cycles of the corresponding $V_{in}$ to encode $n$-bit stochastic numbers. The bit length is determined by $n$.

### Stochastic number correlation

The correlation between the stochastic numbers is quantified using the Pearson correlation $\rho(a, b) = \frac{wz-xy}{\sqrt{(w+x)(w+y)(x+z)(y+z)}}$[26], where $w, x, y$, and $z$ represent the counts of 1-1, 1-0, 0-1, and 0-0 pairs for the two stochastic numbers $a$ and $b$, respectively.

### Edge detection

The Roberts cross operator consists of two $2 \times 2$ kernels, i.e. $\begin{bmatrix} 1 & 0 \\ 0 & -1 \end{bmatrix}$ and $\begin{bmatrix} 0 & 1 \\ -1 & 0 \end{bmatrix}$. As stochastic computing works on the probability domain, each pixel $(i,j)$ of a grayscale image needs to be initially normalised into a probability, denoted as $P(i,j)$. Hence, for a localised pixel map $\begin{bmatrix} P(i,j) & P(i+1,j) \\ P(i,j+1) & P(i+1,j+1) \end{bmatrix}$, each SNE in the stochastic Roberts cross operator is used to encode two positively correlated stochastic numbers, the probabilities of which correspond to the diagonal values in the localised pixel map. As such, two SNEs of the operator encode two pairs of positively correlated stochastic numbers that serve as the inputs to the XOR gates. A pair is input into the XOR gate to yield the $x$ of the output gradient at pixel $(i,j)$, denoted as $|Gx| = |P(i,j) - P(i+1,j+1)|$, while the other pair and XOR gate yield the $y$ component, denoted as $|Gy| = |P(i+1,j) - P(i,j+1)|$. $|Gx|$ and $|Gy|$ are then averaged by the MUX to obtain the absolute magnitude of the approximate gradient. Given the high device uniformity and fabrication yield, the operator is built with two randomly selected memristor based SNEs. The two SNEs exhibit similar $P_{uncorrelated}$-$V_{in}$ relation. No specific or additional memristor calibration, circuit redesign, or testing to obtain the $P_{uncorrelated}$-$V_{in}$ relations are required to implement the SNEs or the stochastic operators. In the edge detection demonstrations, the hardware is used to fully perform the edge extraction, and the following software is used to present and visualise the results. Limited by the scalability of the lab-based realisation of the stochastic Roberts cross operator and parallel signal generation and testing, large-scale edge detection on *The Horse in Motion*, vehicles on highway, and real-time MRI[27] is conducted via simulation in Python3.

## Data availability

The data used in this study are available at Figshare [https://doi.org/10.6084/m9.figshare.28879031].

## Code availability

The code used in this study is available at Code Ocean [https://doi.org/10.24433/CO.7752939.v1].

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

## Acknowledgements

G.H.H. acknowledges support from CUHK (4055227) and RGC (24200521), J.F.P. from RGC (24200521), Y.L. from SHIAE (RNE-p3-21), L.W.T.N. from NTU (9069) and MOE AcRF Tier 1 (10658), and S.G. from National Key Research and Development Program of China (2023YFB3208003).

## Author contributions

L.K.S., P.Y.L., G.H.H. designed the experiments. L.K.S., P.Y.L., J.F.P., Y.L., S.W.L. performed the experiments. L.K.S., P.Y.L., G.H.H. analysed the data. L.K.S., G.H.H. prepared the figures. L.K.S., G.H.H. wrote the manuscript. L.K.S., P.Y.L., J.F.P., Y.L., S.W.L., S.B.W., L.N., T.H., K.P.P., S.G., G.H.H. discussed the results from the experiments and commented on the manuscript.

## Competing interests

The authors declare no competing interests.
