## [Transparent Peer Review file · Nature Communications]

Lightweight error-tolerant edge detection using memristor-enabled stochastic computing

Corresponding Author: Professor Guohua Hu

Version 0:

Reviewer comments:

Reviewer #1

(Remarks to the Author)

In the manuscript entitled "Lightweight, error-tolerant edge detection using memristor-enabled stochastic logics", the authors Song et al. presented stochastic edge detection performed using memristor based stochastic logics. The memristors were fabricated via a scalable solution-based method and then assembled into stochastic logics for performing edge detection tasks, with 95% less computational cost while withstanding 50% bit-flip errors. Overall, the work on stochastic edge detection is very interesting and holds great potential to lead to efficient edge visual hardware and systems. Successful development of such edge visual hardware and systems can impact computer vision applications in autonomous driving, virtual and augmented reality, medical imaging diagnosis, industrial automation, and beyond.

However, before the manuscript can be recommended for publication in Nature Communications, the authors are suggested to address the comments below:

- 1) The stochastic logics are based on solution-processed hBN memristors that exhibit stochastic switching. Stochastic switching seems key for the stochastic logics implementation. How about the other memristors? Can the other memristors be adapted to the stochastic logic circuit designs? Please comment on the versatility of the design of the stochastic logics.
- 2) In terms of the fabrication of the hBN memristors, a thin-layer of SiO₂ is used, and the authors briefly mentioned that this SiO₂ layer could not only improve the fabrication yield but also preserve the stochastic switching characteristics. Please explain the working principle of the memristors with this SiO₂ layer and how this SiO₂ layer can impact the stochastic switching characteristics.
- 3) Following the previous comment, the authors briefly mentioned that the yield of the memristors approximated 100%. However, to be convincing, the authors need to present testing with statistical evidence to support this.
- 4) As presented, a successful implementation of the stochastic logics typically requires multiple memristors to be integrated and function simultaneously. The cycle-to-cycle switching testing as presented in Figure 2 may not be sufficient to assess the suitability of the memristors for implementation of the stochastic logics. Device-to-device testing and stochasticity variations are required for a better assessment.
- 5) Following the previous comment, as presented in Figure 2, the authors performed switching testing of the memristors for 1000 cycles. However, for practical computing applications, 1000 cycles are not sufficient. More rigorous testing of the memristors is required to assess the endurance of the memristors.
- 6) Stochastic number encoders are presented for encoding the input data into stochastic numbers, exploiting the switching stochasticity of the memristors. The stochasticity in memristors has been reported for developing true random number generators for secure cryptography. See Nature communications 12.1 (2021): 2906; Nature communications 8.1 (2017): 882. What would be the key difference between the true random number generators and the stochastic number encoders discussed in this work? Please clarify.

7) Following the previous comment, stochastic number encoders have been recently reported implemented using memristors, for instance, MTJ and Mott memristors. Besides, Quantum devices and integrated circuits have also been reported for stochastic number encoder implementation. Please refer to the refs, e.g. Nature 573.7774 (2019): 390-393; Nature Communications 15.1 (2024): 2812; IEEE Transactions on Computer-Aided Design of Integrated Circuits and Systems 37.12 (2018): 3056-3066. To allow the readers better assess the stochastic number encoders presented in this work, can the authors please compare the stochastic number encoders in this work with the state-of-the-art reports. Please address the key difference and advantages/disadvantages of using the memristors for stochastic number encoder implementation.

8) As shown in Figure 3, the stochastic logics can be configured in different correlations for stochastic number encoding, and the statistical probabilities are highly governed by the correlations. However, it would be difficult to follow the correlations. Please address this with perhaps clearer definition, quantification, and/or visualization.

9) Using the stochastic logics, the authors designed a hardware Roberts cross operator for performing edge detection. The edge detection shows excellent performance in terms of the computational cost and noise tolerance. However, I wonder whether the stochastic logics can be adapted to the other operator designs for other image processing tasks and computing tasks in other domains? See Nature communications 13.1 (2022): 5578; Nature Communications 14.1 (2023): 7199; Nature 573.7774 (2019): 390-393. Please clarify this to improve the overall impact of the work.

10) Following the edge detection demonstration in small scales, the authors presented the potential for real-time, large-scale edge detection for video processing via simulation studies. Practical realization of such edge detection applications would require large-scale integrated circuits and/or computing chips. To improve the overall impact of the work, please comment on the potential and also challenges of integrating the stochastic logics for practical realization of the large-scale integrated circuits and/or computing chips.

(Remarks on code availability)

The code provide enough instructions for installing and running the application.

Reviewer #2

(Remarks to the Author)

This article prepares a memristor using hBN and proposes a stochastic computational method for edge detection. Although the method has some application prospects, I think it is not suitable to be published in nature communications, mainly for the following reasons, the device performance is poor, the threshold voltage is too dispersed, which is not conducive to the application in a hardware system; the size of the system circuit is too small, the system circuit contains only two memristors, and the circuit contains too many gates, and the role of the device has been weakened, and it does not reflect the amnesia in this application. The application scenario used for edge detection is not novel, and it is suggested to be applied in more novel scenarios. To summarize, I think this article does not meet the high level requirements of nature communications, and I suggest to submit it to other articles.

(Remarks on code availability)

The code is only for the computation of edge detection evaluation metrics in the text

Reviewer #3

(Remarks to the Author)

The manuscript by Song et al. discusses implementing stochastic bitstreams using memristive devices and using the stochastic bitstreams for applications such as edge detection. Although the authors presented interesting results, there are serious questions regarding the novelty and technical soundness of the work.

1. The authors presented the techniques as if they were new, using terms such as “we propose ...” for the stochastic bitstream generation and operation. In fact, these techniques were well established. For example, see Knag et al. “A Native Stochastic Computing Architecture Enabled by Memristors,” in IEEE Transactions on Nanotechnology, vol. 13, no. 2, pp. 283-293, 2014, for stochastic bitstream generation and operation discussions, and Gaba et al. “Stochastic memristive devices for computing and neuromorphic applications” in Nanoscale, 5, 5872-5878, 2013 for device implementations. Curiously, these original papers were not included, and novelty over the existing techniques was not discussed.

2. A major challenge of using stochastic switching to generate the bitstreams is the device wearout, since memristive devices have limited write endurance. In this work a 100 bit bitstream is used to encode one number for one operation. However, the endurance is only 120,000 cycles, meaning only 1200 operations can be achieved during the lifetime of the device. This is clearly impractical. Longer endurance has to be demonstrated to show advances over prior studies.

3. Similarly, another challenge of switching-based bitstream operation is the high energy during device switching. The authors did not present any energy discussions using the measured data. Please include measured energy results from the edge detection operations, and benchmark against efficient digital implementations.

4. Additionally, the authors use the limited retention to relax the device back to the reset state after each pulse. This will be a very slow process. Please add discussions on the speed of the operations (with 100 bits, for example, which provides sufficient precision), and compare with efficient digital implementations.

5. Finally, it's not clear how many devices the authors tested, and how repeatable the $V_{\text{threshold}}$ is for these devices. Distribution of $V_{\text{threshold}}$ should be presented. If $V_{\text{threshold}}$ needs to be tuned for every device, this approach is not

practical.

(Remarks on code availability)

Version 1:

Reviewer comments:

Reviewer #1

(Remarks to the Author)

The authors thoroughly revised the previous manuscript based on my concerns and suggestions accordingly. Now, I recommend this work to accept for publication as is.

(Remarks on code availability)

Reviewer #2

(Remarks to the Author)

After thoroughly reviewing the authors' response and the revised manuscript, I am not convinced that the major concerns raised in my initial review have been adequately addressed. While the proposed memristor-based system using hBN shows some potential, some core issues remain unresolved. The hardware system presented remains limited in its scale, consisting of only two devices, with excessive reliance on additional circuit components. This diminishes the actual role of the memristors in the edge detection task, making it difficult to see how the proposed approach can fully leverage the benefits of the hardware design. The edge detection application itself, as mentioned earlier, is not particularly novel, and the system does not provide a complete hardware-based solution. For example, although the circuit produces pulse outputs to represent pixel intensity, further software processing is still required to complete the edge detection task. This leaves the hardware's contribution relatively small in comparison to what is needed for a full system. I would encourage the authors to explore more innovative application scenarios and refine the hardware implementation to better demonstrate its advantages. Therefore, I maintain my recommendation that this manuscript is not suitable for publication in Nature Communications at this stage.

(Remarks on code availability)

Reviewer #4

(Remarks to the Author)

See attachment.

After checking the rebuttal letter, it seems to me the previous questions have not been addressed in full.

The authors has conducted more rigorous study on device endurance, which can exceed 5×10^6 cycles and outperforms the current studies. However, this is still insufficient to support practical applications, and the paper needs to provide an explanation on this point. The paper's assessment of latency and energy consumption is not sufficiently convincing. The evaluation uses data from components on the PCB, but the data obtained in this way does not reflect the situation of the actual chip. Because the clock period of digital circuits can be very fast (\sim GHz) in real chips, but the switching time of the memristor in the paper is only 50ns (20MHz), so the clock periods of the two should not be equivalent. In addition, and are required to be adjusted according to the input, and other components (such as DACs) are needed to give the voltages, and this part of the overhead cannot be ignored. Also, the authors only provide a comparison with other memristor-enabled implementations, lacking a comparison with efficient digital implementations. Another point is it cannot be seen from Figure R1 that no additional device calibration or circuit design is required. For example, the V_{th} and V_{hold} of device 4 and device 5 differ by more than 0.2V.

(Remarks on code availability)

Version 3:

Reviewer comments:

Reviewer #1

(Remarks to the Author)

I believe that this revised manuscript has been further improved by successfully addressing the reviewers' suggestions and concerns. Therefore, I think it can be recommended to accept for publication without further updates.

(Remarks on code availability)

Reviewer #4

(Remarks to the Author)

The authors have addressed my questions. I have no other comments.

(Remarks on code availability)

Point-by-point response to reviewers' comments

Reviewer #1:

In the manuscript entitled “Lightweight, error-tolerant edge detection using memristor-enabled stochastic logics”, the authors Song et al. presented stochastic edge detection performed using memristor based stochastic logics. The memristors were fabricated via a scalable solution-based method and then assembled into stochastic logics for performing edge detection tasks, with 95% less computational cost while withstanding 50% bit-flip errors. Overall, the work on stochastic edge detection is very interesting and holds great potential to lead to efficient edge visual hardware and systems. Successful development of such edge visual hardware and systems can impact computer vision applications in autonomous driving, virtual and augmented reality, medical imaging diagnosis, industrial automation, and beyond.

However, before the manuscript can be recommended for publication in Nature Communications, the authors are suggested to address the comments below:

1) The stochastic logics are based on solution-processed hBN memristors that exhibit stochastic switching. Stochastic switching seems key for the stochastic logics implementation. How about the other memristors? Can the other memristors be adapted to the stochastic logic circuit designs? Please comment on the versatility of the design of the stochastic logics.

We thank the reviewer for this insightful comment.

Indeed, stochastic switching is key for the stochastic logic implementation, and our hBN stochastic memristors are one of the approaches to implement the stochastic logics. Other memristors (and also other types of devices) may also be suitable for the implementation (including the stochastic number encoders and stochastic logics) as long as they exhibit reliable stochasticity for stochastic number encoding with controllable probabilities and correlations.

More detailedly, the stochastic number encoders provide the stochasticity for the stochastic logics. Therefore, the focus on the stochastic logic implementation lies in the design of the stochastic number encoders. The early designs of stochastic number encoders were based on electronic circuits, for instance, using *linear feedback shift registers (LFSR)* that could convert pseudo-random numbers into stochastic numbers (*Ichihara et al., IEEE Transactions on Emerging Topics in Computing 7.1 (2016): 31-43*). As the memristor technology develops, memristor-enabled encoders were then proposed. Arising from the underlying switching mechanisms, memristors typically exhibit an inherent switching stochasticity. For instance, electrochemical metallization memristors can exhibit a switching stochasticity due to the stochastic ion diffusion (e.g. *Dutta et al., Nature Communications 13.1 (2022): 2571*); magnetic tunnel junction based memristors can exhibit a switching stochasticity due to thermal instability of the magnetic tunnel junctions (e.g. *Borders et al., Nature 573.7774 (2019): 390-393*). Exploiting the inherent switching stochasticity, the memristor-enabled stochastic number encoders can

directly encode stochastic numbers. For example, the aforementioned two studies on electrochemical metallization memristors and magnetic tunnel junction memristors were reported to implement stochastic number encoders and logics. Compared to the early designs, memristor-enabled stochastic number encoders and logics are more lightweight and can save the computational cost.

In response to the reviewer's comment and to give the audience a broader image on stochastic number encoder and logic designs, we have now included the following discussion in the revised manuscript:

(Line 82) SNEs are units encoding the data into stochastic numbers, and have been conventionally realised using electronic circuits by harnessing the stochasticity present in the circuits (13–15). As the memristor technology develops, memristors-based SNEs were proposed. Memristors often exhibit a stochasticity in switching, originating from the underlying switching mechanisms. For example, due to the stochastic diffusion of the conductive elements, filamentary memristors switch with an intrinsic stochasticity (16). This characteristic makes memristors promising for realising stochastic computing, including compact SNE design and implementation (17–19). Figure 2a,...

2) In terms of the fabrication of the hBN memristors, a thin-layer of SiO₂ is used, and the authors briefly mentioned that this SiO₂ layer could not only improve the fabrication yield but also preserve the stochastic switching characteristics. Please explain the working principle of the memristors with this SiO₂ layer and how this SiO₂ layer can impact the stochastic switching characteristics.

We thank the reviewer for this comment.

For electrochemical metallization memristors with the switching based on the metallic filament formation and rupture, oxides such as SiO_x and TiO_x are widely used as the dielectric medium for the diffusion of the metal cations, as reported in state-of-the-art studies (e.g. *Yang et al., Nature nanotechnology* 8.1 (2013): 13-24; *Sun et al., Advanced Functional Materials* 24.36 (2014): 5679-5686; *Ding et al., IEEE Transactions on Electron Devices* 69.3 (2022): 1034-1040). In these devices, the diffusion of the metal cations and formation of conductive metallic filaments through the oxides lead to the switching of the memristors.

In our work, we incorporate a 10 nm SiO_x layer in the memristors to mainly minimize the wash-off of the deposited hBN during the following photolithographic patterning process, thereby increasing the fabrication yield (see our response in the following comment). During memristor operation, we expect that the silver ions could first diffuse through the thin SiO_x layer and then the hBN layer underneath to form conductive metallic filaments and switch the memristors. Due to the thickness difference between the SiO_x layer (10 nm) and the hBN layer (~260 nm), the diffusion of the silver ions through the hBN layer governs the switching behaviour of the memristors. More specifically, in this diffusion process, the silver filaments are expected to reach an equilibrium yet metastable state driven by the voltage bias and the accompanying Joule heat (e.g. *Wang et al., Nature materials* 16.1 (2017): 101-

108; Hsiung *et al.*, *ACS nano* 4.9 (2010): 5414-5420; Tang *et al.*, *Advanced Materials* 31.49 (2019): 1902761). This leads to the volatile switching behaviour with a self-reset threshold switching characteristic in our memristors. Besides, the diffusion of the silver ions in the hBN layer is essentially stochastic following a stochastic *Ornstein-Uhlenbeck* process (Dutta *et al.*, *Nature Communications* 13.1 (2022): 2571). This leads to the switching stochasticity in our memristors.

Therefore, in response to the reviewer's comment, the 10 nm SiO_x layer is incorporated to increase the fabrication yield, while the switching behaviour of our hBN memristors is essentially governed by the silver ion diffusion in the hBN layer.

To clarify, we have now included the following change in the revised manuscript:

(Line 98) Briefly, hBN is produced by liquid-phase exfoliation (Fig. S1) and used to fabricate the memristors in a Pt/Au/hBN/SiO_x/Ag configuration (Fig. S2). **This solution-based fabrication approach is scalable, with high yield and uniformity (20).** We show in Fig. 2b and c an array of the fabricated memristors and the structure of a typical device. **As demonstrated in Fig. S3, the success rate of the array fabrication via a random sampling test is 100%.**

(Line 315) **10 nm SiO_x minimises wash-off of the deposited hBN during the photolithographic patterning process and thereby increases the yield.** The device substrate is Si/SiO₂. The slot-die coater is Ossila L2005A1. The evaporator is IVS EB-600. During device fabrication, the hBN layer after deposition is baked at 200°C for 2 hours. **For a typical memristor, given the thickness difference between the SiO_x (10 nm) and hBN (~260 nm) layers, the switching behaviour is governed by the formation and rupture of silver filaments in the hBN layer.**

3) Following the previous comment, the authors briefly mentioned that the yield of the memristors approximated 100%. However, to be convincing, the authors need to present testing with statistical evidence to support this.

We thank the reviewer for this comment.

To study the yield of the memristor fabrication, we randomly select and test 10 devices from an array of the memristors, as shown in Fig. R1a, b. For each of the memristor devices, the device is tested for 20 cycles. Figure R1c plots the histograms of the hold voltage (V_{hold}) and threshold voltage (V_{th}) for the 10 memristor devices. This random sampling test shows a fabrication yield approximating 100%. However, we note that further rigorous test must be carried out to verify a 100% success rate in large-scale manufacturing. To clarify, we have now included Fig. R1 as Fig. S3 in the revised SI.

Figure R1. Device-to-device stochasticity test. (a) 12×12 memristor array in a crossbar configuration, with a fabrication yield approximating 100% (replotted from Fig. 2b), and (b) the corresponding schematic array showing the devices randomly selected for the sampling test. (c) Distributions of the measured V_{hold} and V_{th} of sampled devices for 20 sweeping cycles, along with the corresponding Gaussian fittings, indicating minimal device-to-device stochasticity – V_{hold} variation 6.6%, V_{th} variation 7.4%. The device-to-devices variation are defined using the standard deviations of the mean V_{hold} (V_{th}) values.

4) As presented, a successful implementation of the stochastic logics typically requires multiple memristors to be integrated and function simultaneously. The cycle-to-cycle switching testing as presented in Figure 2 may not be sufficient to assess the suitability of the memristors for implementation of the stochastic logics. Device-to-device testing and stochasticity variations are required for a better assessment.

We thank the reviewer for this comment. Yes, indeed, a uniform device-to-device stochasticity is required for the successful implementation of the stochastic logics.

As shown in Fig. R1, we test and study the device-to-device stochasticity through the distributions of the hold voltage (V_{hold}) and threshold voltage (V_{th}) of the 10 randomly sampled devices. It can be observed that the device-to-device stochasticity is minimal, with a variation of 6.6% in V_{hold} , and 7.4% in V_{th} . Note the device-to-device variations are defined using the standard deviations of the mean V_{hold} (V_{th}) values. This uniformity in the device-to-device stochasticity allows us to implement the stochastic logics without excess further device calibrations or circuit designs.

To clarify, as discussed, we have included Fig. R1 as Fig. S3 in the revised SI; we have also made the following discussion in the revised manuscript:

(Line 115) We have also tested the device-to-device stochasticity (Fig. S3), proving a high uniformity, with variation of 6.6% in V_{hold} and 7.4% in V_{th} . This uniformity, along with the high fabrication yield, allows for SNE implementation without excess device calibrations or circuit reconfigurations.

5) Following the previous comment, as presented in Figure 2, the authors performed switching testing of the memristors for 1000 cycles. However, for practical computing applications, 1000 cycles are not sufficient. More rigorous testing of the memristors is required to assess the endurance of the memristors.

We thank the reviewer for this comment.

To begin with, we clarify that the switching behaviour as presented in Fig. 2 is conducted with 1,000 full sweeping cycles. The full sweeping cycle test is conducted to better demonstrate the stochastic yet stable volatile switching characteristic of the memristors. Indeed, an endurance of 1,000 full sweeping cycles is not sufficient for practical applications. However, in practical applications, for instance the stochastic logics in our work, the memristors usually operate in a pulsed mode, that is, the memristors are applied with pulsed signals instead of full sweeping cycles. Therefore, besides the full sweeping cycle test, we have also performed test using pulsed signals to study the endurance. As presented in Fig. S4 in the original SI, we proved an endurance exceeding 120,000 cycles.

In this revision, to rigorously study the endurance of our memristors, we have performed test using pulsed signals exceeding 5×10^6 cycles, as shown in Fig. R2. The rigorous endurance test proves a highly stable operation of our memristors, and the endurance outperforms the current studies, e.g. 10^6 cycles in *Cheong et al., Nature Communications 15.1 (2024): 6318*; 10^6 cycles in *Woo et al., Nature Communications 15.1 (2024): 3245*; 600 cycles in *Teja et al., Nature Communications 15.1 (2024): 2334*.

Figure R2. Endurance test. Endurance test of a typical memristor undergoing 5×10^6 consecutive test cycles under pulsed stimuli. For each test cycle, a $20 \mu\text{s}$ voltage pulse of 10 V is set to fully program the memristor and an $80 \mu\text{s}$ voltage pulse of 0.1 V is set to read the output. The output of the memristor is amplified by an operational amplifier and measured by an oscilloscope. The high (i.e. off) and low (i.e. on) resistance states in each test cycle are computed based on the oscilloscope measurement, and plotted in blue and red, respectively. Both the states remain stable throughout the full test.

To clarify, we have now replaced the original Fig. S4 (now Fig. S6) with Fig. R2 in the revised SI, and made the following change in the revised manuscript:

(Line 123) Indeed, the endurance test for over 5×10^6 cycles proves a highly stable yet stochastic switching of our memristors (Fig. S6).

6) Stochastic number encoders are presented for encoding the input data into stochastic numbers, exploiting the switching stochasticity of the memristors. The stochasticity in memristors has been reported for developing true random number generators for secure cryptography. See Nature communications 12.1 (2021): 2906; Nature communications 8.1 (2017): 882. What would be the key difference between the true random number generators and the stochastic number encoders discussed in this work? Please clarify.

We thank the reviewer for this comment. Yes, indeed, as commented by the reviewer, memristors have been reported in true random number generation exploiting their inherent stochasticity.

Stochastic numbers share certain common properties with true random numbers, such as having identical bit weight, following a Bernoulli process, and exhibiting low correlations (*Jeavons et al., IEEE Transactions on Information Theory 40.3 (1994): 716-720*). However, they do present two key differences:

- Stochastic numbers have to encode the information in a string of stochastic bits to represent a probability between 0 to 1, whereas the true random numbers are a string of random bits of 0s and 1s that must be equal and unbiased (i.e. representing a probability of 0.5 for both 0s and 1s bits) to ensure true randomness (*Clark et al., IEEE Transactions on VLSI Systems 26.10 (2018): 2027-2037; Wei et al., Optics letters 34.12 (2009): 1876-1878*);
- Stochastic numbers do not need to strictly adhere to the cryptographic requirements in terms of the patterns and correlations of the 0s and 1s bits. In contrast, true random numbers must guarantee no patterns or correlations to pass the cryptographic tests. Importantly, as discussed, we note that the focus of our work is exploiting the well-regulated probability and correlations of the stochastic numbers to facilitate the stochastic logics and computing applications.

7) Following the previous comment, stochastic number encoders have been recently reported implemented using memristors, for instance, MTJ and Mott memristors. Besides, Quantum devices and integrated circuits have also been reported for stochastic number encoder implementation. Please refer to the refs, e.g. Nature 573.7774 (2019): 390-393; Nature Communications 15.1 (2024): 2812; IEEE Transactions on Computer-Aided Design of Integrated Circuits and Systems 37.12 (2018): 3056-3066. To allow the readers better assess the stochastic number encoders presented in this work, can the authors please compare the

stochastic number encoders in this work with the state-of-the-art reports. Please address the key difference and advantages/disadvantages of using the memristors for stochastic number encoder implementation.

We thank the reviewer for this very important comment.

Yes, indeed, other memristors and also the other types of devices and circuits have been reported in stochastic number encoder development exploiting their inherent stochasticity. To better assess our stochastic number encoders, we compare them with those reported in state-of-the-art studies in the key features, including the encoding speed, endurance, and circuit complexity, as presented in Table R1:

- **Encoding speed:** Our stochastic number encoders can encode information with a speed of 455 kbit/s, outperforming most encoders based on the other memristors. The encoding speed of our stochastic number encoders is governed by the switching mechanism of our memristors, i.e. the diffusion of the silver ions. However, note the general encoding speed of memristor-enabled encoders is not yet comparable to the LFSR-based encoders developed by advanced CMOS processes. Further materials and device engineering of memristors may be adopted to enhance the encoding speed further by mitigating the switching and relaxation times (Teja et al., Nature Communications 15.1 (2024): 2334).
- **Endurance:** Using our memristors, our stochastic number encoders can allow for an excellent endurance ($>5 \times 10^6$ cycles under pulsed stimuli). This outperforms the other memristor-enabled encoders. However, we note that the endurance may still be behind the LFSR-based encoders (though no practical endurance data is available in our survey).
- **Circuit complexity:** Our stochastic number encoders, leveraging the volatile switching and intrinsic stochasticity of the hBN memristors, present advantages in hardware and computational cost over the LFSR-based encoders, requiring three orders of magnitude fewer electrical components.

Table R1. Comparison between the stochastic number encoders implemented using LFSR, magnetic tunnel junction (MTJ), Mott memristors, filamentary memristors, and our memristors. Data in the Table is reported in [1] Borders et al., Nature 573.7774 (2019): 390-393; [2] Knag et al., IEEE Transactions on Nanotechnology 13.2 (2014): 283-293; [3] Seo et al., Advanced Materials (2024): 2402490; [4] Rhee et al., Nature Communications 14.1 (2023): 7199; [5] Deng et al., IEEE Electron Device Letters 44.10 (2023): 1776 – 1779 ; [6] Woo et al., Nature Communications 13.1 (2022): 5762.

	Max encoding speed (kbit/s)	Endurance (cycles)	Transistor/memristor count
LFSR [1]	10^7	/	1194
LFSR [2]	10^5	/	/
MTJ [1]	10^4	/	4
Mott [3]	400	$>1.05 \times 10^5$	/
Mott [4]	263~3,846	/	/
Mott [5]	/	5×10^4	3
Filamentary [6]	1	10^6	3

To clarify and allow the audience to better assess our stochastic number encoders, we have now included Table R1 as Table S2 in the revised SI and the following discussion in the revised manuscript:

(Line 82) SNEs are units encoding the data into stochastic numbers, and have been conventionally realised using electronic circuits by harnessing the stochasticity present in the circuits (13–15). As the memristor technology develops, memristors-based SNEs were proposed. Memristors often exhibit a stochasticity in switching, originating from the underlying switching mechanisms. For example, due to the stochastic diffusion of the conductive elements, filamentary memristors switch with an intrinsic stochasticity (16). This characteristic makes memristors promising for realising stochastic computing, including compact SNE design and implementation (17–19). Figure 2a...

(Line 105) See also Fig. S4 for the ultrafast volatile switching (switching time ~50 ns, and relaxation time ~1,200 ns) and the ultralow energy consumption (~33 fJ per bit) of the memristors. Due to the stochastic diffusion of the silver ions, the switching exhibits stochasticity in both V_{th} and V_{hold} . The volatile switching eliminates the need for any peripheral circuits or excessive resetting for SNE implementation and operation, while the switching stochasticity can be harnessed for stochastic number encoding, leading to compact circuit designs of SNEs.

8) As shown in Figure 3, the stochastic logics can be configured in different correlations for stochastic number encoding, and the statistical probabilities are highly governed by the correlations. However, it would be difficult to follow the correlations. Please address this with perhaps clearer definition, quantification, and/or visualization.

We thank the reviewer for this very important comment.

The correlation between the stochastic numbers can be quantified using the Pearson correlation (*Alaghi et al., 2013 IEEE 31st International Conference on Computer Design (ICCD). IEEE, 2013*), following the definition of Pearson correlation coefficient,

$$\rho(a, b) = \frac{wz - xy}{\sqrt{(w+x)(w+y)(x+z)(y+z)}}$$

where w, x, y, and z represent the counts of 1-1, 1-0, 0-1, and 0-0 pairs for the two stochastic numbers a and b, respectively.

Using the Pearson correlation coefficient, we conduct correlation analysis on the stochastic number inputs of the stochastic logics in Fig. 3. The results, as shown in Fig. R3, demonstrate pairwise correlations between the inputs in the uncorrelated, positively correlated, and negatively correlated conditions. This confirms that the stochastic logics can work in the desired correlation conditions and perform the corresponding logic operations as summarized in Table 1 in the manuscript.

Figure R3. Pearson correlation coefficients of the stochastic logics. Pairwise Pearson correlation of the inputs a, b, and s (if MUX) of the stochastic logics in the uncorrelation and correlation conditions. The Pearson correlation coefficient is defined as $\rho(a, b) = \frac{wz - xy}{\sqrt{(w+x)(w+y)(x+z)(y+z)}}$, where w, x, y, and z represent the counts of 1-1, 1-0, 0-1, and 0-0 pairs for the two stochastic numbers a and b, respectively.

To address the comment of the reviewer and allow the audience to better assess the correlation of our stochastic logics, we have now included Fig. R3 as Fig. S10 in the revised SI and the following discussion in the revised manuscript:

(Line 180) We present in Fig. S10 the pairwise correlations between the inputs of the above stochastic logics in the uncorrelated, positively correlated and negatively correlated conditions, and summarise the statistical relations between $P(a)$, $P(b)$ and $P(c)$ in Table S1. The pairwise correlations and the statistical relations confirm that our stochastic logics can work in the desired correlation conditions and conduct the corresponding logic operations for performing edge detection tasks. *Pearson correlation* is adopted here to quantify the correlations (see Methods). Note that...

(Line 329) **Stochastic number correlation:** The correlation between the stochastic numbers is quantified using the *Pearson correlation* (21), following,

$$\rho(a, b) = \frac{wz - xy}{\sqrt{(w+x)(w+y)(x+z)(y+z)}}$$

where w, x, y, and z represent the counts of 1-1, 1-0, 0-1, and 0-0 pairs for the two stochastic numbers a and b, respectively.

9) Using the stochastic logics, the authors designed a hardware Roberts cross operator for performing edge detection. The edge detection shows excellent performance in terms of the computational cost and noise tolerance. However, I wonder whether the stochastic logics can be

adapted to the other operator designs for other image processing tasks and computing tasks in other domains? See Nature communications 13.1 (2022): 5578; Nature Communications 14.1 (2023): 7199; Nature 573.7774 (2019): 390-393. Please clarify this to improve the overall impact of the work.

We thank the reviewer for this important comment.

As reported in state-of-the-art studies, stochastic logics can be used as fundamental logic units for performing various tasks, including Bayesian networks, hardware acceleration, optimization, and image processing (*Zheng et al., Nature communications 13.1 (2022): 5578; Aadit et al., Nature Electronics 5.7 (2022): 460-468; Borders et al., Nature 573.7774 (2019): 390-393; Zhao et al., 2019 IEEE International Electron Devices Meeting (IEDM). IEEE, 2019*). In our work, we investigate the use of the stochastic logics in image processing applications.

Beyond image processing applications, we are currently exploring the feasibility of using the stochastic logics in Bayes inference and multimodal fusion for efficient, error-tolerant decision-making. Here we present some preliminary results in Fig. R4. Figure R4a illustrates the circuit design of two basic topologies in Bayes networks using the stochastic MUX. Figure R4b shows the circuit design of Bayes inference using the stochastic logics. Figure R4c presents the simulation results of stochastic logic-enabled Bayes multimodal fusion on a video frame for efficient objection detection in autonomous driving. Hardware implementation results are being optimized and will be ready for discussion in our following work.

Figure R4. Stochastic logics for Bayes inference and multimodal fusion. (a) Circuit design of two basic topologies, i.e. *one-parent-one-child* and *two-parent-one-child*, in Bayes networks using the stochastic MUX. (b) Circuit design of Bayes inference using the stochastic logics. (c) Simulation results of stochastic logic-enabled Bayes multimodal fusion on a video frame for objection detection with improved confidence in autonomous driving.

10) Following the edge detection demonstration in small scales, the authors presented the potential for real-time, large-scale edge detection for video processing via simulation studies. Practical realization of such edge detection applications would require large-scale integrated circuits and/or computing chips. To improve the overall impact of the work, please comment on the potential and also challenges of integrating the stochastic logics for practical realization of the large-scale integrated circuits and/or computing chips.

We thank the reviewer for this important comment.

- **Advantages and potential for large-scale stochastic edge detection tasks**

As shown in Table R1, our stochastic number encoders have demonstrated advantageous performance in terms of the encoding speed (~ 455 kbit/s), energy consumption (~ 33 fJ per bit), and the required count of electrical components (3 transistor/memristor count).

The advantageous performance allows for a high stochastic number encoding speed, easily exceeding 100 kbit/s. This speed ensures the independence of the consecutive bits, following the Bernoulli

process. Assuming 100-bit stochastic numbers (though 4-bit are sufficient as demonstrated in Fig. 4) are used for information encoding, performing the edge detection tasks as presented in large-scale can in principle achieve a frame rate of 1,000 fps, outperforming the requirement of the practical image processing scenarios including autonomous driving, e.g. 30-45 fps in advanced driver assistance systems (Daniel et al., *Nature* 629.8014 (2024): 1034-1040).

Also, the advantages allow for edge detection at low computational cost. It is estimated that the stochastic operator can consume ~95% less energy with 4-bit stochastic number inputs. Similarly, with 64-bit stochastic number inputs, it can still consume ~90% less energy. [Please see the detailed energy consumption estimation in our below response to the 3rd comment of Reviewer 3.] Therefore, this lightweight stochastic edge detection approach, if up-scaled, holds a great promise to realize practical computing hardware that is highly suitable for deployment in edge visual applications where computational resources (including the computational hardware and power) are constrained. The approach can extract early visual cues including contours and textures, facilitating timely decision-making.

Though our stochastic edge detection in large scale can in principle achieve fast frame rates and low computational cost, circuit designs that encode information in parallel using multiple devices may be explored to enhance the efficiency in a large-scale implementation (Knag et al., *IEEE Transactions on Nanotechnology* 13.2 (2014): 283-293). In this approach, multiple bits may be encoded at each timestep, thereby improving the encoding speed. On the other hand, as shown in Fig. R5, although the switching time of the memristors is fast (~50 ns) due to the rapid formation of conductive filaments, the relaxation process is relatively slow (~1,200 ns). To mitigate the relaxation time, further materials and device engineering may be needed. For instance, employing thinner/different dielectrics and introducing faster switching mechanisms may be viable strategies (Teja et al., *Nature Communications* 15.1 (2024): 2334). These optimizations can potentially increase the encoding speed to MHz and even GHz, in alignment with the current advanced integrated circuit and chip technologies for large-scale integrated circuit design and fabrication for real-time stochastic edge detection tasks.

Figure R5. Switching speed and energy consumption. (a) Electrical response to 1 μs pulsed signal, showing (b) a switching time of ~ 50 ns, a switching energy consumption of ~ 33 fJ, and a relaxation time of $\sim 1,200$ ns.

- *Challenges for large-scale stochastic edge detection tasks*

Indeed, as commented by the reviewer, our current hardware implementation and stochastic edge detection at the laboratory level is in small scales, limited to the challenges in scaling up the memristor fabrication. However, we believe the small-scale demonstration can be readily scaled up to large-scale integrated circuits for real-time stochastic edge detection tasks, given the high yield and device-to-device stochasticity uniformity of the memristor fabrication method.

Large-scale implementation of the integrated circuits requires efforts for addressing the large-scale peripheral circuit design and fabrication, integration of the memristors with the peripheral circuits, and parallel operation of the large-scale circuits. Besides, a large-scale implementation can introduce non-idealities, for instance, noises and delays from the devices and circuits. Further efforts in hardware-algorithm codesigns have to be made to deal with the non-idealities. For instance, computational algorithms may be tailored to accommodate the non-idealities.

In terms of the large-scale integrated circuit manufacturing level, *Process, Voltage, and Temperature (PVT) analysis* may be performed to evaluate the performance and reliability of the large-scale circuits, taking into account of the variations in the process technology, supply voltage, and operating temperature that can impact the overall performance and functionality of the circuits. In practical cases, PVT analysis may be conducted via modelling and simulation techniques – process models capture the statistical variations in the process parameters, which can be then incorporated into the circuit-level simulations; voltage and temperature variations can be typically modelled using worst-case or statistical analyses. Through PVT analysis, it is possible to optimize the circuit design, improve the yield, and enhance the reliability of the large-scale integrated circuits for real-time stochastic edge detection tasks.

In response to the reviewer’s comment and to give the audience a broader image on the scalability of our stochastic logics, we have now included the following discussion in the revised manuscript:

(Line 291) Given the remarkable edge detection performance, the scalability of the memristors, and the compactness of the circuit designs, our stochastic logics are readily scaled-up and generalised towards the development of lightweight, error-tolerant edge visual hardware and systems. **Arising from the ultrafast switching of the memristors, a large-scale stochastic logic circuit with 100-bit stochastic number encoding can in principle easily achieve edge detection with a frame rate of over 1,000 fps, fulfilling the requirements of applications ranging from autonomous driving and virtual/augmented reality to industrial automation and medical imaging diagnosis.** To explore the feasibility, we show via simulation the effectiveness of large-scale stochastic logic circuits for video edge detection (Supplementary Video 1-3). **Though promising, large-scale stochastic logic circuit implementation requires efforts for realising large-scale design and fabrication of the memristors and peripheral circuits,**

integration of the memristors with the peripheral circuits, and parallel operation of the large-scale circuits. Hardware and algorithm codesigns may need to be implemented to address or accommodate the non-idealities, e.g. errors, delays, and drifts from the devices and circuits. Besides, the performance and reliability of large-scale manufacturing must be evaluated, given that the variations in the process technology, supply voltage, and operating temperature can impact the overall performance and functionality of the circuits significantly.

Reviewer #2:

This article prepares a memristor using hBN and proposes a stochastic computational method for edge detection. Although the method has some application prospects, I think it is not suitable to be published in nature communications, mainly for the following reasons, the device performance is poor, the threshold voltage is too dispersed, which is not conducive to the application in a hardware system; the size of the system circuit is too small, the system circuit contains only two **amnesia, and the circuit contains too many gates, and the role of the device has been weakened, and it does not reflect the **amnesia** in this application. The application scenario used for edge detection is not novel, and it is suggested to be applied in more novel scenarios.**

To summarize, I think this article does not meet the high level requirements of nature communications, and I suggest to submit it to other articles.

We thank the reviewer for the comment, and we feel sorry that the reviewer found our initial submission was not suitable for publication in Nature Communications.

Before we address the reviewer's comment, we need to point out '**amnesia**' is not technically correct. **It is a medical term for loss of memory caused by brain damage or brain diseases.** However, in neuromorphic computing, memristors are the technical term to describe the devices with memristive behaviour. More specifically, **volatile memristors** are the term to describe **our hBN memristors with the short-term memory.**

Nevertheless, we find the comment of the reviewer is not well supported, and we believe all the concerns of the reviewer are fully addressed in this revision, based on our following point-by-point response:

- ***Excellent device performance conducive to hardware application***

Our hBN memristors demonstrate excellent device performance. The key device performance includes a stable volatile switching with self-reset thresholding characteristics (Fig. 2), a prolonged and highly stable switching endurance ($>5 \times 10^6$ cycles; Fig. R2), an ultrafast switching (switching time ~ 50 ns, and relaxation time $\sim 1,200$ ns; Fig. R5), and an ultralow switching energy consumption (~ 33 fJ per bit; Fig. R5). To the best knowledge of us, this device performance outperforms or at least is competitive to the state-of-the-art advances.

Figure R2. Endurance test. Endurance test of a typical memristor undergoing 5×10^6 consecutive test cycles under pulsed stimuli. For each test cycle, a 20 μs voltage pulse of 10 V is set to fully program the memristor and an 80 μs voltage pulse of 0.1 V is set to read the output. The output of the memristor is amplified by an operational amplifier and measured by an oscilloscope. The high (i.e. off) and low (i.e. on) resistance states in each test cycle are computed based on the oscilloscope measurement, and plotted in blue and red, respectively. Both the states remain stable throughout the full test.

Figure R5. Switching speed and energy consumption. (a) Electrical response to 1 μs pulsed signal, showing (b) a switching time of ~50 ns, a switching energy consumption of ~33 fJ, and a relaxation time of ~1,200 ns.

The excellent device performance is for sure conducive to the hardware application, i.e. the stochastic number encoders and logics in our work. As shown in Table R1, our stochastic number encoders have demonstrated advantageous performance in terms of the encoding speed (~455 kbit/s), endurance ($>5 \times 10^6$ cycles), and the counts of required electrical components (3 transistor/memristor count).

- **Encoding speed:** Our stochastic number encoders can encode information with a speed

of 455 kbit/s, outperforming most encoders based on the other memristors. The encoding speed of our stochastic number encoders is governed by the switching mechanism of our memristors, i.e. the diffusion of the silver ions. However, note the general encoding speed of memristor-enabled encoders is not yet comparable to the LFSR-based encoders developed by advanced CMOS processes. Further materials and device engineering of memristors may be adopted to enhance the encoding speed further by mitigating the switching and relaxation times (*Teja et al., Nature Communications 15.1 (2024): 2334*).

- **Endurance:** Using our memristors, our stochastic number encoders can allow for an excellent endurance ($>5 \times 10^6$ cycles under pulsed stimuli). This outperforms the other memristor-enabled encoders. However, we note that the endurance may still be behind the LFSR-based encoders (though no practical endurance data is available in our survey).
- **Circuit size and complexity:** Our stochastic number encoders, leveraging the volatile switching and intrinsic stochasticity of the hBN memristors, present advantages in hardware and computational cost over the LFSR-based encoders, requiring three orders of magnitude fewer electrical components.

Table R1. Comparison between the stochastic number encoders implemented using LFSR, magnetic tunnel junction (MTJ), Mott memristors, filamentary memristors, and our memristors. Data in the Table is reported in [1] *Borders et al., Nature 573.7774 (2019): 390-393*; [2] *Knag et al., IEEE Transactions on Nanotechnology 13.2 (2014): 283-293*; [3] *Seo et al., Advanced Materials (2024): 2402490*; [4] *Rhee et al., Nature Communications 14.1 (2023): 7199*; [5] *Deng et al., IEEE Electron Device Letters 44.10 (2023): 1776 – 1779*; [6] *Woo et al., Nature Communications 13.1 (2022): 5762*.

	Max encoding speed (kbit/s)	Endurance (cycles)	Transistor/ memristor count
LFSR [1]	10^7	/	1194
LFSR [2]	10^5	/	/
MTJ [1]	10^4	/	4
Mott [3]	400	$>1.05 \times 10^5$	/
Mott [4]	263~3,846	/	/
Mott [5]	/	5×10^4	3
Filamentary [6]	1	10^6	3
This work	455	$>5 \times 10^6$	3

Importantly, in practical applications, our stochastic number encoders and logics can allow for a high stochastic number encoding speed with well-regulated probabilities and correlations leveraging the advantageous performance. For instance, the information encoding speed can easily exceed 100 kbit/s, while maintaining the independence of the consecutive bits, following the Bernoulli process. Assuming 100-bit stochastic number bit length (though 4-bit is sufficient as demonstrated in Fig. 4) is used for information encoding, performing the edge detection tasks as presented in our work in large-scales can in principle achieve a frame rate of 1,000 fps, outperforming the requirements of practical image processing scenarios including autonomous

driving, e.g. 30-45 fps in advanced driver assistance systems (Daniel et al., Nature 629.8014 (2024): 1034-1040).

- **Reliable stochasticity conducive to hardware application**

Our hBN memristors demonstrate well dispersed yet reliable stochasticity. As shown in Fig. 2, both the hold voltage (V_{hold}) and threshold voltage (V_{th}) of our hBN memristors prove a well dispersed stochasticity, with both V_{hold} (0.23 ± 0.18 V) and V_{th} (0.78 ± 0.39 V) well fitting Gaussian distributions. As discussed, this stochasticity inherent in the memristor switching arises from the stochastic diffusion of the silver ions. Governed by the reliable stochastic diffusion process of the silver ions, the switching stochasticity in principle is a reliable process. To investigate this further, as demonstrated in Fig. S5, the switching stochasticity can well fit an *Ornstein-Uhlenbeck* process, suggesting the stability of the stochasticity of our memristors in prolonged switching operations. Indeed, as shown in Fig. R2, our memristors demonstrate a highly stable switching endurance ($>5 \times 10^6$ cycles).

Besides the well dispersity of the stochasticity of single devices, we have performed rigorous device-to-device stochasticity assessment of our memristors. We randomly select and test 10 devices from an array of the memristors, as shown in Fig. R1a, b. For each of the memristor devices, the device is tested for 20 cycles. Figure R1c plots the histograms of V_{hold} and V_{th} for the 10 devices. This sampling test shows a fabrication yield of 100%. More importantly, the test shows a uniform device-to-device stochasticity.

We test the device-to-device stochasticity through the distributions of V_{hold} and V_{th} of the 10 devices. It can be observed that the device-to-device stochasticity is uniform, with variation of 6.6% in V_{hold} , and 7.4% in V_{th} . The device-to-device variations are defined using the standard deviations of the mean V_{hold} (V_{th}) values. This uniformity in the device-to-device stochasticity, in addition to the high fabrication yield, allows us to implement the stochastic logics without excess further device calibrations or circuit reconfigurations.

Figure R1. Device-to-device stochasticity test. (a) 12×12 memristor array in a crossbar configuration, with a fabrication yield approximating 100% (replotted from Fig. 2b), and (b)

the corresponding schematic array showing the devices randomly selected for the sampling test. (c) Distributions of the measured V_{hold} and V_{th} of sampled devices for 20 sweeping cycles, along with the corresponding Gaussian fittings, indicating minimal device-to-device stochasticity – V_{hold} variation 6.6%, V_{th} variation 7.4%. The device-to-device variations are defined using the standard deviations of the mean V_{hold} (V_{th}) values.

The reliable stochasticity with the well dispersity and uniform device-to-device stochasticity again is for sure conducive to the hardware application –

- A reliable stochasticity allows for prolonged and highly stable switching endurance, critical for the hardware applications, i.e. the stochastic number encoders and logics in our work;
 - A well dispersity ensures that the stochastic number encoders and logics can perform information encoding and computing with well-regulated probabilities and correlations;
 - Uniform device-to-device stochasticity guarantees that multiple memristor devices can be integrated with logic circuits for the hardware implementation of stochastic number encoders and logics, laying the foundation of stochastic edge detection application in our work.
- ***Compact circuit size leading to minimal hardware and computational cost***

As discussed, our stochastic number encoders leveraging the volatile switching and intrinsic stochasticity of the hBN memristors lead to the highly compact circuit designs, outperforming the other memristors-based encoders and LFSR-based encoders (Table R1). Particularly, compared to the LFSR-based encoders, **our stochastic number encoders require three orders of magnitude fewer electrical components**. The much more compact circuit size of our stochastic number encoders not only leads to much lower hardware and computational cost but also much less energy consumption.

To discuss this further, here we estimate the energy consumption for both our stochastic and the conventional edge detection operators. We note that the estimation is conducted under the same computational precision, i.e. each n-bit binary number is represented by 2^n -bit stochastic numbers.

The energy consumption of our stochastic edge detection operators can be mainly contributed to 1) the memristors and 2) the remaining comparators and logic gates:

- ***Memristors***: The total energy of a pulsed voltage signal can be determined by integrating the product of the transient voltage and the current over time, as defined by, $E = \int_0^t V(t)I(t)dt$. Given that the switching process involves Joule heat, this energy can be segregated into the switching energy and the excess energy (Teja et al., *Nature Communications* 15.1 (2024): 2334), as shown in Fig. R5. The switching energy is from the resistive switching process, while the excess energy represents the energy dissipated

throughout the device after the completion of the switching process. As shown in Fig. R5, the switching energy is estimated as ~ 33 fJ per bit. Encoding a 2^n -bit stochastic number consumes $\sim 33 \times 2^n$ fJ.

- **Comparators and logic gates:** The energy consumption of the remaining comparators and logic gates for edge detection depends on the specific hardware implementation. Here we estimate the energy consumption based on the required counts of logic gates and clock cycles for our stochastic edge detection operator, following $W = k \times T_c \times P$, where T_c represents the clock cycle, k is the required counts of T_c , and P is the total power of all the logic gates. For our stochastic Roberts cross operator (Fig. R6a), 2^n -bit stochastic numbers are processed serially. Therefore, it requires $(2^n + 1)T_c$ and thus, $(2^n + 1)T_c P_{stochastic}$ energy. Assume the edge detection is conducted at a frequency of 100 kHz, T_c is 10 μ s. To calculate $P_{stochastic}$, we refer to the product power datasheet of the logic gate chips used in our work (Table R2).

As such, the total energy consumption of our stochastic Roberts cross operator can be estimated as the sum of the energy consumption of the memristors and the remaining circuit components.

To perform edge detection in the binary computing domain, the energy consumption is incurred by the conventional edge detection operators. Here we estimate the energy consumption of the conventional Roberts cross operator (Fig. R6b). As illustrated, the conventional Roberts cross operator consists of two n -bit subtractors and one n -bit adder. Each n -bit subtractor and adder can be built using n full adders (FA) and several XOR gates, as illustrated in Fig. R6c. Considering parallel computation, each n -bit subtractor and adder requires $(2n + 3)T_c$. Therefore, assuming two subtractors run in parallel, the conventional Roberts cross operator requires $2(2n + 3)T_c$ and thus, $2(2n + 3)T_c P_{conventional}$ energy. Similarly, we assume $T_c = 10 \mu$ s and refer to Table R2 to calculate $P_{conventional}$. Here we note that binary computing does not encode the binary n -bit input into stochastic numbers, and additional circuits and computational operations are often necessitated to deal with the errors.

We present in Fig. R6d-f the performance comparison between our stochastic and the conventional Roberts cross operators in terms of required clock cycles and energy consumption for edge detection. As mentioned, the comparison is conducted under the same computational precision. As shown in Fig. R6d, the stochastic operator requires fewer clock cycles to complete edge detection with the stochastic number inputs < 16 bits. However, as the bit length increases beyond 16 bits, the stochastic operator requires exponentially increased clock cycles, surpassing the conventional counterpart. Nevertheless, as presented in Fig. R6e, the stochastic operator can maintain a lower energy consumption within 2,048 bits. To provide a more intuitive representation of the energy efficiency, we plot the energy consumption ratio of the stochastic operator to the conventional counterpart in Fig. R6f. The results show that **with 4-bit stochastic number inputs, the stochastic operator can consume $\sim 95\%$ less energy.** Similarly, with 64-bit stochastic number inputs, it can still consume $\sim 90\%$ less energy.

Table R2. Power consumption of logic gate chips. The data are from the product datasheets.

	Chip	Power (mW)	Channel	Power/channel (mW)
AND	SN74HC08N	120.0	4	30.0
OR	HD74LS32P	25.7	4	6.4
XOR	HD74LS86P	32.0	4	8.0
MUX	SN74LS157N	96.0	4	24.0
Comparator	LM393	14.4	2	7.2

Figure R6. Performance and energy consumption comparison between the stochastic and conventional Roberts cross operators. Circuit designs of (a) our stochastic and (b) the conventional Roberts cross operator. The 2^n -bit inputs to our stochastic Roberts operator are stochastic numbers. (c) Circuit design of a n -bit subtractor and adder in (b). Relation of required (d) clock cycles and (e) energy consumption with respect to the bit length of the binary numbers and stochastic numbers. (f) Energy consumption ratio in (e). When the length of the stochastic numbers is within 2,048 bits, the energy consumption of the stochastic Roberts operator is lower. All comparisons are conducted at a frequency of 100 kHz for demonstration purposes.

- **Up-scalability of the stochastic logics for large-scale stochastic edge detection**

Our current hardware implementation is in small scales, limited to the challenges in scaling up the memristor fabrication and integration at the research laboratory level. With this limitation, our work focuses on the hardware implementation of individual stochastic number encoders and stochastic logics. However, as demonstrated, the simplified logic operations and error tolerance of stochastic logics have proved their suitability for stochastic edge detection applications. To demonstrate feasibility, we implement a lightweight edge detection operator

rather than a complete system, and use the lightweight edge detection operator for image processing. Edge detection is a fundamental function that requires continuous operation to extract early visual cues for subsequent complex computing tasks.

However, we believe **the small-scale demonstration can be readily scaled up to large-scale integrated circuits** for real-time stochastic edge detection tasks, given 1) the high yield and device-to-device stochasticity uniformity of our hBN memristor fabrication method, and 2) the readily adaptability and up-scalability of the circuit designs.

Large-scale implementation of the integrated circuits requires efforts for addressing the large-scale peripheral circuit design and fabrication, integration of the memristors with the peripheral circuits, and parallel operation of the large-scale circuits. The large-scale implementation can introduce non-idealities, e.g., errors, noises, delays, and/or drifts from the devices and circuits. Further efforts in hardware-algorithm codesigns have to be made to deal with the non-idealities. For instance, computational algorithms may be tailored to accommodate the non-idealities.

In terms of the large-scale integrated circuit manufacturing, *Process, Voltage, and Temperature (PVT) analysis* may be performed to evaluate the performance and reliability of the large-scale circuits, considering the variations in the process technology, supply voltage, and operating temperature that can impact the overall performance and functionality of the circuits. In practical cases, PVT analysis may be conducted via modelling and simulation techniques – process models capture the statistical variations in the process parameters, which can be then incorporated into the circuit-level simulations; voltage and temperature variations can be typically modelled using worst-case or statistical analyses. Through PVT analysis, it is possible to optimize the circuit design, improve the yield, and enhance the reliability of the large-scale integrated circuits for real-time stochastic edge detection tasks.

- ***Critical role of the memristors to enable stochastic computing***

As we discuss in the manuscript, hardware implementation of stochastic computing requires 1) stochastic number encoders for information encoding into stochastic numbers (Fig. 2), and 2) stochastic logics integrating the stochastic number encoders for stochastic number processing (Fig. 3). Our hBN memristors play a critical role in the hardware implementation of the stochastic number encoders and logics – by harnessing the intrinsic stochasticity from our hBN memristors, the stochastic number encoders and logics can allow information encoding and processing with well-regulated probabilities and correlations.

More importantly, as discussed above, the excellent performance of the memristors, and the reliable stochasticity of the memristors with a good dispersity and uniform device-to-device variation are highly conducive to the hardware application of the stochastic number encoders and logics, enabling fast information encoding speed, minimal energy consumption, excellent endurance, compact circuit size and complexity, and minimized hardware and computational cost.

Therefore, we believe the role of our hBN memristors is not weakened at all but rather serves as the critical core of the hardware application.

- ***Novelty of the edge detection application***

As we discuss in the manuscript, extracting key image features to enable efficient user-scene interaction and decision-making is a key technology enabling the wide range of computer vision applications, ranging from autonomous driving, virtual and augmented reality, medical imaging diagnosis, to industrial automation, and beyond. However, successful key image feature extracting has been a challenging topic due to the intensive computational workload, particularly for the edges with constrained computational resources.

In this context, edge detection is widely employed as a fundamental image pre-processing technique to extract the key early visual cues, such as the shallow features of color, contour and texture, for efficient image understanding and initial decision-making. **As edge detection serves as the backbone for image processing tasks, edge detection has always been at the forefront of research**, with focus on improving the computing efficiency and robustness towards the noises and errors for practical applications (*Cetinkaya et al, Proceedings of the IEEE/CVF Conference on Computer Vision and Pattern Recognition. 2024; Zhou et al., Proceedings of the IEEE/CVF Conference on Computer Vision and Pattern Recognition. 2024; Yang et al., IEEE Transactions on Industrial Informatics. 2024*).

We understand that novelty is not merely defined as whether the research has been done before, but whether this research can solve practical problems. Given the fundamental importance of edge detection for image processing and the wide range of applications, **we consider our research on edge detection is novel, and we believe our research findings can lead to impacts to the field** towards the development of efficient edge visual hardware and systems.

Beyond edge detection, however, we do explore other potential applications with our stochastic number encoders and logics, with the efforts to address practical problems in the field. For instance, we are currently exploring the feasibility of using the stochastic logics in Bayes inference and multimodal fusion for efficient, error-tolerant decision-making. Here we present preliminary results in Fig. R4. Figure R4a illustrates the circuit design of two basic topologies in Bayes networks using the stochastic MUX. Figure R4b shows the circuit design of Bayes inference using the stochastic logics. Figure R4c presents the simulation results of stochastic logic-enabled Bayes multimodal fusion on a video frame for efficient objection detection in autonomous driving. Hardware implementation results are being optimized and will be ready for discussion in our following work.

Figure R4. Stochastic logics for Bayes inference and multimodal fusion. (a) Circuit design of two basic topologies, i.e. *one-parent-one-child* and *two-parent-one-child*, in Bayes networks using the stochastic MUX. (b) Circuit design of Bayes inference using the stochastic logics. (c) Simulation results of stochastic logic-enabled Bayes multimodal fusion on a video frame for objection detection with improved confidence in autonomous driving.

Reviewer #3:

The manuscript by Song et al. discusses implementing stochastic bitstreams using memristive devices and using the stochastic bitstreams for applications such as edge detection. Although the authors presented interesting results, there are serious questions regarding the novelty and technical soundness of the work.

1. The authors presented the techniques as if they were new, using terms such as “we propose ...” for the stochastic bitstream generation and operation. In fact, these techniques were well established. For example, see Knag et al "A Native Stochastic Computing Architecture Enabled by Memristors," in IEEE Transactions on Nanotechnology, vol. 13, no. 2, pp. 283-293, 2014, for stochastic bitstream generation and operation discussions, and Gaba et al. “Stochastic memristive devices for computing and neuromorphic applications” in Nanoscale, 5, 5872-5878, 2013 for device implementations. Curiously, these original papers were not included, and novelty over the existing techniques was not discussed.

We thank the reviewer for this important comment.

Yes, as commented by the reviewer, these two early papers provide the groundbreaking perspectives and contributions to the field of stochastic computing: 1) The first paper proposed a native stochastic computing architecture based on non-volatile memristors, with discussions on the strategy to program stochastic bitstream, and machine learning application demonstration through simulation; 2) The second paper presented hardware implementation of memristor-enabled stochastic number generators with regulated probability and demonstrated a multi-bit synapse using a 4-memristor array for neuromorphic computing. When we were preparing our investigation, we had mainly focused on state-of-the-art edge detection advances via the stochastic computing approach, and might have therefore overlooked some of the early studies in the field. To give the audience a better image of the stochastic computing field, we have now cited the early papers in this revision.

To highlight the novelty of our work, we discuss the key innovations and differences from the two early studies (and also the prior studies in general) as follows:

- ***Stochastic logics design***: In our hardware implementation of stochastic logics, to eliminate the need for excessive reset operations and peripheral circuits, we design stochastic logics using volatile stochastic memristors with a self-reset threshold switching. As demonstrated, the design sufficiently minimizes the hardware cost for the implementation, thereby reducing computational cost and energy consumption for the operation of the stochastic logics. In these two early studies and other prior studies, non-volatile memristors were typically used. As such, excessive reset operations and peripheral circuits were required for the stochastic number generation operations.
- ***Stochastic number encoding***: In our demonstration, the stochasticity inherent in the threshold switching voltage (V_{th}) of our memristors is exploited directly using a simple circuit design to perform the stochastic number encoding. This allows for a convenient and efficient information

encoding into stochastic numbers, arising from the fast-switching speed of the memristors (switching time ~ 50 ns and relaxation time $\sim 1,200$ ns as shown in Fig. R5). For instance, the stochastic number encoding speed can easily exceed 100 kbit/s; Assuming 100-bit stochastic numbers (though 4-bit are sufficient as demonstrated in Fig. 4) are used for information encoding, performing the edge detection tasks as presented in large-scale can in principle achieve a frame rate of 1,000 fps, outperforming the requirement of the practical image processing scenarios including autonomous driving, e.g. 30-45 fps in advanced driver assistance systems (Daniel et al., *Nature* 629.8014 (2024): 1034-1040). In these two early studies and other prior studies, the stochasticity might be derived from the switching time of the memristors. This can be challenging and lead to excessive hardware and algorithm operations and cost.

- **Stochastic number correlation:** In our implementation of stochastic logics and stochastic edge detection, the stochastic numbers present well-regulated correlations. As we demonstrate, the correlation can be employed as an approach to facilitate lightweight and error-tolerant hardware design and stochastic edge detection. In these two early studies and other prior studies, however, the correlation was typically considered a non-ideal factor and thus, techniques were adopted to ensure the uncorrelation of stochastic numbers, including re-randomization, regeneration, and isolation (Tehrani et al., *IEEE Transactions on Signal Processing* 58.9 (2010): 4883-4896; Ting et al., 2016 *IEEE 34th International Conference on Computer Design (ICCD)*. *IEEE*, 2016). This can lead to expense of increased circuit complexity and computational cost.
- **Application scenarios:** In our work, using the stochastic logics, we focus on lightweight error-tolerant image processing. As discussed, in edge computer vision, enabling efficient user-scene interaction and decision-making has been challenging, as a result of intensive computational workload and the constrained computational resources at edges. Our demonstration, with minimal hardware and computational cost to achieve error-tolerant edge detection, underscores the potential in developing efficient edge visual hardware and systems for autonomous driving, virtual and augmented reality, medical imaging diagnosis, industrial automation, and beyond. We believe this highlights the novelty and the impact of our work.

Therefore, in a short summary, we believe our work has demonstrated the novelty in terms of the stochastic logics design, stochastic number encoding, stochastic number correlation, and application scenarios. In response to the reviewer's comment and to give the audience a broader image on our stochastic logic designs, we have now cited the early papers in the field and have made the following changes in the revised manuscript:

(Line 82) SNEs are units encoding the data into stochastic numbers, and have been conventionally realised using electronic circuits by harnessing the stochasticity present in the circuits (13–15). As the memristor technology develops, memristors-based SNEs were proposed. Memristors often exhibit a stochasticity in switching, originating from the underlying switching mechanisms. For example, due to the stochastic diffusion of the conductive elements, filamentary memristors switch with an intrinsic stochasticity (16). This characteristic makes memristors promising for realising stochastic computing, including compact SNE design and implementation (17–19). Figure 2a...

(Line 108) The volatile switching eliminates the need for any peripheral circuits or excessive resetting for SNE implementation and operation, while the switching stochasticity can be harnessed for stochastic number encoding, leading to compact circuit designs of SNEs.

(Line 180) We present in Fig. S10 the pairwise correlations between the inputs of the above stochastic logics in the uncorrelated, positively correlated and negatively correlated conditions, and summarise the statistical relations between $P(a)$, $P(b)$ and $P(c)$ in Table S1. The pairwise correlations and the statistical relations confirm that our stochastic logics can work in the desired correlation conditions and conduct the corresponding logic operations for performing edge detection tasks. *Pearson correlation* is adopted here to quantify the correlations (see Methods). Note that ...

Line 239-279 on the detailed performance comparison between our stochastic and the conventional edge detection operators, highlighting that our stochastic edge detection approach can consume 95% less energy compared to the conventional binary computing approach.

(Line 293) Arising from the ultrafast switching of the memristors, a large-scale stochastic logic circuit with 100-bit stochastic number encoding can in principle easily achieve edge detection with a frame rate of over 1,000 fps, fulfilling the requirements of applications ranging from autonomous driving and virtual/augmented reality to industrial automation and medical imaging diagnosis. To explore...

2. A major challenge of using stochastic switching to generate the bitstreams is the device wearout, since memristive devices have limited write endurance. In this work a 100 bit bitstream is used to encode one number for one operation. However, the endurance is only 120,000 cycles, meaning only 1200 operations can be achieved during the lifetime of the device. This is clearly impractical. Longer endurance has to be demonstrated to show advances over prior studies.

We thank the reviewer for this important comment.

Indeed, as commented by the reviewer, an endurance of 120,000 cycles could not rigorously prove a reliable switching property of our memristors and was not sufficient for practical applications. In this revision, to rigorously study the endurance of our memristors, we have performed test using pulsed signals exceeding 5×10^6 cycles, as shown in Fig. R2. The rigorous endurance test proves a highly stable operation of our memristors, and the endurance outperforms the current studies, e.g. 10^6 cycles in *Cheong et al., Nature Communications 15.1 (2024): 6318*; 10^6 cycles in *Woo et al., Nature Communications 15.1 (2024): 3245*; 600 cycles in *Teja et al., Nature Communications 15.1 (2024): 2334*.

Figure R2. Endurance test. Endurance test of a typical memristor undergoing 5×10^6 consecutive test cycles under pulsed stimuli. For each test cycle, a 20 μs voltage pulse of 10 V is set to fully program the memristor and an 80 μs voltage pulse of 0.1 V is set to read the output. The output of the memristor is amplified by an operational amplifier and measured by an oscilloscope. The high (i.e. off) and low (i.e. on) resistance states in each test cycle are computed based on the oscilloscope measurement, and plotted in blue and red, respectively. Both the states remain stable throughout the full test.

To clarify, we have now replaced the original Fig. S4 (now Fig. S6) with Fig. R2 in the revised SI, and made the following change in the revised manuscript:

(Line 123) Indeed, the endurance test for over 5×10^6 cycles proves a highly stable yet stochastic switching of our memristors (Fig. S6).

3. Similarly, another challenge of switching-based bitstream operation is the high energy during device switching. The authors did not present any energy discussions using the measured data. Please include measured energy results from the edge detection operations, and benchmark against efficient digital implementations.

We thank the reviewer for this very important comment.

Here we estimate the energy consumption for both our stochastic and the conventional edge detection operators. We note that the estimation is conducted under the same computational precision, i.e. each n -bit binary number is represented by 2^n -bit stochastic numbers.

The energy consumption of our stochastic edge detection operators can be mainly contributed to 1) the memristors and 2) the remaining comparators and logic gates :

- **Memristors:** The total energy of a pulsed voltage signal can be determined by integrating the product of the transient voltage and the current over time, as defined by, $E = \int_0^t V(t)I(t)dt$.

Given that the switching process involves Joule heat, this energy can be segregated into the switching energy and the excess energy (Teja et al., *Nature Communications* 15.1 (2024): 2334), as shown in Fig. R5. The switching energy is from the resistive switching process, while the excess energy represents the energy dissipated throughout the device after the completion of the switching process. As shown in Fig. R5, the switching energy is estimated as ~ 33 fJ per bit. Encoding a 2^n -bit stochastic number consumes $\sim 33 \times 2^n$ fJ.

Figure R5. Switching speed and energy consumption. (a) Electrical response to 1 μ s pulsed signal, showing (b) a switching time of ~ 50 ns, a switching energy consumption of ~ 33 fJ, and a relaxation time of $\sim 1,200$ ns.

- **Comparators and logic gates:** The energy consumption of the remaining comparators and logic gates for edge detection, however, depends on the specific hardware implementation. This was the reason that we did not discuss the energy consumption in our original manuscript.

Here we estimate the energy consumption based on the required counts of logic gates and clock cycles for our stochastic edge detection operator, following $W = k \times T_c \times P$, where T_c is the clock cycle, k is the required counts of T_c , and P is the total power of all the logic gates. For our stochastic Roberts cross operator (Fig. R6a), 2^n -bit stochastic numbers are processed serially. Therefore, it requires $(2^n + 1)T_c$ and thus, $(2^n + 1)T_c P_{stochastic}$ energy. Assume the edge detection is conducted at a frequency of 100 kHz, T_c is 10 μ s. To calculate $P_{stochastic}$, we refer to the product power datasheet of the logic gate chips used in our work (Table R2).

As such, the total energy consumption of our stochastic Roberts cross operator can be estimated as the sum of the energy consumption of the memristors and the remaining circuit components.

To perform edge detection in the binary computing domain, the energy consumption is incurred by the conventional edge detection operators. Here we estimate the energy consumption of the conventional Roberts cross operator (Fig. R6b). As illustrated, the conventional Roberts cross operator consists of

two n-bit subtractors and one n-bit adder. Each n-bit subtractor and adder can be built using n full adders (FA) and several XOR gates, as illustrated in Fig. R6c. Considering parallel computation, each n-bit subtractor and adder requires $(2n + 3)T_c$. Therefore, assuming two subtractors run in parallel, the conventional Roberts cross operator requires $2(2n + 3)T_c$ and thus, $2(2n + 3)T_c P_{conventional}$ energy. Similarly, we assume $T_c = 10 \mu s$ and refer to Table R2 to calculate $P_{conventional}$. Here we note that binary computing does not encode the binary n-bit input into stochastic numbers, and additional circuits and computational operations are often necessitated to deal with the errors.

We present in Fig. R6d-f the performance comparison between our stochastic and the conventional Roberts cross operators in terms of required clock cycles and energy consumption for edge detection. As mentioned, the comparison is conducted under the same computational precision. As shown in Fig. R6d, the stochastic operator requires fewer clock cycles to complete edge detection with the stochastic number inputs <16 bits. However, as the bit length increases beyond 16 bits, the stochastic operator requires exponentially increased clock cycles, surpassing the conventional counterpart. Nevertheless, as presented in Fig. R6e, the stochastic operator can maintain a lower energy consumption within 2,048 bits. To provide a more intuitive representation of the energy efficiency, we plot the energy consumption ratio of the stochastic operator to the conventional counterpart in Fig. R6f. The results show that **with 4-bit stochastic number inputs, the stochastic operator can consume ~95% less energy**. Similarly, with 64-bit stochastic number inputs, it can still consume ~90% less energy.

Table R2. Power consumption of logic gate chips. The data are from the product datasheets.

	Chip	Power (mW)	Channel	Power/channel (mW)
AND	SN74HC08N	120.0	4	30.0
OR	HD74LS32P	25.7	4	6.4
XOR	HD74LS86P	32.0	4	8.0
MUX	SN74LS157N	96.0	4	24.0
Comparator	LM393	14.4	2	7.2

Figure R6. Performance and energy consumption comparison between the stochastic and conventional Roberts cross operators. Circuit designs of (a) our stochastic and (b) the conventional Roberts cross operator. The 2^n -bit inputs to our stochastic Roberts operator are stochastic numbers. (c) Circuit design of a n-bit subtractor and adder in (b). Relation of required (d) clock cycles and (e) energy consumption with respect to the bit length of the binary numbers and stochastic numbers. (f) Energy consumption ratio in (e). When the length of the stochastic numbers is within 2,048 bits, the energy consumption of the stochastic Roberts operator is lower. All comparisons are conducted at a frequency of 100 kHz for demonstration purposes.

To clarify and give the audience a detailed image on the energy consumption of our stochastic logics, we have now included Fig. R6 as Fig. 6 in the manuscript, with the estimation in Line 245-285.

4. Additionally, the authors use the limited retention to relax the device back to the reset state after each pulse. This will be a very slow process. Please add discussions on the speed of the operations (with 100 bits, for example, which provides sufficient precision), and compare with efficient digital implementations.

We thank the reviewer for this very important comment.

To investigate the relaxation time of our memristors, we plot in Fig. R5 the electrical response of a typical memristor under 1 μ s pulsed voltage signal. As shown, the memristor exhibits a rapid switching (~ 50 ns) and relaxation ($\sim 1,200$ ns). As such, for information encoding, it takes down to ~ 2.2 μ s to encode each bit. This means a stochastic number encoding speed of ~ 455 kbit/s, outperforming the other memristor-enabled stochastic number encoders in state-of-the-art studies (Table R1).

Table R1. Comparison between the stochastic number encoders implemented using LFSR, magnetic tunnel junction (MTJ), Mott memristors, filamentary memristors, and our memristors. Data in the Table is reported in [1] Borders *et al.*, *Nature* 573.7774 (2019): 390-393; [2] Knag *et al.*, *IEEE Transactions on Nanotechnology* 13.2 (2014): 283-293; [3] Seo *et al.*, *Advanced Materials* (2024): 2402490; [4] Rhee *et al.*, *Nature Communications* 14.1 (2023): 7199; [5] Deng *et al.*, *IEEE Electron Device Letters* 44.10 (2023): 1776 – 1779 ; [6] Woo *et al.*, *Nature Communications* 13.1 (2022): 5762.

	Max encoding speed (kbit/s)	Endurance (cycles)	Transistor/memristor count
LFSR [1]	10^7	/	1194
LFSR [2]	10^5	/	/
MTJ [1]	10^4	/	4
Mott [3]	400	$>1.05 \times 10^5$	/
Mott [4]	263~3,846	/	/
Mott [5]	/	5×10^4	3
Filamentary [6]	1	10^6	3
This work	455	$>5 \times 10^6$	3

The advantageous performance allows for a high stochastic number encoding speed in applications, easily exceeding 100 kbit/s. This speed ensures the independence of the consecutive bits, following the Bernoulli process. Assuming 100-bit stochastic numbers (although 4-bit are sufficient as demonstrated in Fig. 4) are used for information encoding, performing the edge detection tasks as presented in large-scale can in principle achieve a frame rate of 1,000 fps, outperforming the practical image processing scenarios including autonomous driving, e.g. 30-45 fps in advanced driver assistance systems (Daniel *et al.*, *Nature* 629.8014 (2024): 1034-1040).

Though our stochastic edge detection in large scale can in principle achieve fast frame rates and low computational cost, circuit designs that encode information with multiple devices in parallel may be explored to enhance efficiency in large-scale implementation (Knag *et al.*, *IEEE Transactions on Nanotechnology* 13.2 (2014): 283-293). In this approach, multiple bits can be encoded at each timestep, thereby improving the encoding speed. On the other hand, as shown in Fig. R5, although the switching time of the memristors is fast (~50 ns) due to the rapid formation of conductive filaments, the relaxation process is relatively slow (~1,200 ns). To mitigate the relaxation time, further device engineering may be needed. For instance, employing a thinner dielectric medium and introducing a faster switching mechanism would be a viable strategy (Teja *et al.*, *Nature Communications* 15.1 (2024): 2334). These optimizations can potentially increase the encoding speed to MHz and even GHz, in alignment with the current advanced integrated circuit and chip technologies for large-scale integrated circuits design and fabrication for real-time stochastic edge detection tasks.

To clarify, we have now included Fig. R5 as Fig. S4 in the revised SI, and have made the following discussion in the revised manuscript:

(Line 105) See also Fig. S4 for the ultrafast volatile switching (switching time ~ 50 ns, and relaxation time $\sim 1,200$ ns) and the ultralow energy consumption (~ 33 fJ per bit) of the memristors. Due to the stochastic diffusion of the silver ions, the switching exhibits stochasticity in both V_{th} and V_{hold} . The volatile switching eliminates the need for any peripheral circuits or excessive resetting for SNE implementation and operation, while the switching stochasticity can be harnessed for stochastic number encoding, leading to compact circuit designs of SNEs.

(Line 293) Arising from the ultrafast switching of the memristors, a large-scale stochastic logic circuit with 100-bit stochastic number encoding can in principle easily achieve edge detection with a frame rate of over 1,000 fps, fulfilling the requirements of applications ranging from autonomous driving and virtual/augmented reality to industrial automation and medical imaging diagnosis. To explore...

5. Finally, it's not clear how many devices the authors tested, and how repeatable the $V_{threshold}$ is for these devices. Distribution of $V_{threshold}$ should be presented. If $V_{threshold}$ needs to be tuned for every device, this approach is not practical.

We thank the reviewer for this very important comment.

In response to the reviewer's comment, we have now performed rigorous assessment of our memristors. We randomly select and test 10 devices from an array of the memristors, as shown in Fig. R1a, b. For each of the memristor devices, the device is tested for 20 cycles. Figure R1c plots the histograms of the hold voltage (V_{hold}) and threshold voltage (V_{th}) for the ten memristor devices. This random sampling test of the device array shows a fabrication yield approximates 100%. However, we note that further rigorous test must be carried out to verify a 100% success rate in large-scale manufacturing.

We further test the device-to-device stochasticity through the distributions of the hold voltage (V_{hold}) and threshold voltage (V_{th}) of the 10 randomly sampled devices. It can be observed that the device-to-device stochasticity is minimal, with a variation of 6.6% in V_{hold} and 7.4% in V_{th} . The device-to-device variation is defined using the standard deviation of the mean V_{hold} (V_{th}) values. This uniformity in the device-to-device stochasticity, in addition to the high fabrication yield, allows us to implement the stochastic logics without excess further device calibrations or circuit designs.

Figure R1. Device-to-device stochasticity test. (a) 12×12 memristor array in a crossbar configuration, with a fabrication yield approximating 100% (replotted from Fig. 2b), and (b) the corresponding schematic array showing the devices randomly selected for the sampling test. (c) Distributions of the measured V_{hold} and V_{th} of sampled devices for 20 sweeping cycles, along with the corresponding Gaussian fittings, indicating minimal device-to-device stochasticity – V_{hold} variation 6.6%, V_{th} variation 7.4%. The device-to-device variations are defined using the standard deviations of the mean V_{hold} (V_{th}) values.

To clarify, we have now included Fig. R1 as Fig. S3 in the revised SI, and have also made the following discussion in the revised manuscript:

(Line 115) We have also tested the device-to-device stochasticity (Fig. S3), proving a high uniformity, with variation of 6.6% in V_{hold} and 7.4% in V_{th} . This uniformity, along with the high fabrication yield, allows for SNE implementation without excess device calibrations or circuit reconfigurations. To evaluate the stability...

Point-by-point response to reviewers' comments

Reviewer #1:

The authors thoroughly revised the previous manuscript based on my concerns and suggestions accordingly. Now, I recommend this work to accept for publication as is.

We thank the reviewer for recommendation of publication in Nature Communications.

Reviewer #2:

After thoroughly reviewing the authors' response and the revised manuscript, I am not convinced that the major concerns raised in my initial review have been adequately addressed. While the proposed memristor-based system using hBN shows some potential, some core issues remain unresolved. The hardware system presented remains limited in its scale, consisting of only two devices, with excessive reliance on additional circuit components. This diminishes the actual role of the memristors in the edge detection task, making it difficult to see how the proposed approach can fully leverage the benefits of the hardware design. The edge detection application itself, as mentioned earlier, is not particularly novel, and the system does not provide a complete hardware-based solution. For example, although the circuit produces pulse outputs to represent pixel intensity, further software processing is still required to complete the edge detection task. This leaves the hardware's contribution relatively small in comparison to what is needed for a full system. I would encourage the authors to explore more innovative application scenarios and refine the hardware implementation to better demonstrate its advantages. Therefore, I maintain my recommendation that this manuscript is not suitable for publication in Nature Communications at this stage.

We appreciate the reviewer for this comment, and we feel sorry that the reviewer still finds our revised manuscript is not yet well fit for publication in *Nature Communications* after we address the reviewer's concerns towards the performance, stability, and variations, etc of the memristors in our revised manuscript.

From this comment, we understand that the reviewer is mainly concerned with 1) the scale of the stochastic hardware circuits/operators, 2) the memristor role in stochastic computing, 3) the novelty of edge detection, and 4) contribution of the hardware-based solution of edge detection. We now address the concerns with the below point-by-point responses:

1) The scale of our stochastic hardware circuits/operators

Our work focuses on memristor-based stochastic computing. The implementation requires the realization of stochastic number encoders and logic circuits. To demonstrate the feasibility, we design and implement memristor-based circuits and a lightweight hardware operator, rather than large-scale hardware computing systems, as proof-of-concept to perform edge detection. With the proof-of-concept demonstration, we further study the feasibility of large-scale stochastic computing and edge detection via simulations (Supplementary Video 1-3).

Indeed, we acknowledge that the scalability of our hardware stochastic circuits and computing is far from comparable to commercial computing chips. We would like to highlight that **this is limited by the scalabilities in device fabrication, circuit integration, and parallel signal processing in lab-based research. Therefore, current studies in neuromorphic computing remain largely at conceptual explorations via simulations.**

In this context, we want to highlight further in detail the small-scale design of our hardware stochastic circuits and computing demonstration is justified:

1.1) The small scale of edge detection operator

Edge detection operators (mainly the convolutional ones) rely on local pixels and their relations to extract visual features/cues in a lightweight manner (e.g. *Bishop et al., Pattern recognition and machine learning. Vol. 4. No. 4. New York: springer, 2006*). These edge detection operators are typically small in their scalability, e.g. 2×2 and 3×3 pixels. As such, building an edge detection operator does not really have to require a large scalability or too many devices. In this work, we adopt the commonly used *Roberts cross* operator that has a scalability of 2×2 pixels.

1.2) The lightweight design of the hardware *Roberts cross* operator using the stochasticity of memristors

Besides the small scale of edge detection operators, the lightweight design of our hardware stochastic *Roberts cross* operator is attributed to the exploration of the intrinsic stochasticity of the memristors. By exploiting the stochasticity of memristors, memristor-based stochastic number encoders (including our work) require three orders of magnitude fewer electrical components compared to those conventionally realised with electronic circuits (e.g. those based on linear feedback shift registers) (Table S1).

Table S1. Comparison between the stochastic number encoders implemented using linear feedback shift registers (LFSR), magnetic tunnel junction (MTJ), Mott memristors, filamentary memristors, and our memristors. Data in the Table is reported in [1] Borders et al., Nature 573.7774 (2019): 390-393; [2] Knag et al., IEEE Transactions on Nanotechnology 13.2 (2014): 283-293; [3] Seo et al., Advanced Materials (2024): 2402490; [4] Rhee et al., Nature Communications 14.1 (2023): 7199; [5] Deng et al., IEEE Electron Device Letters 44.10 (2023): 1776 – 1779; [6] Woo et al., Nature Communications 13.1 (2022): 5762.

	Max encoding speed (kbit/s)	Endurance (cycles)	Transistor/ memristor count
LFSR [1]	10^7	/	1194
LFSR [2]	10^5	/	/
MTJ [1]	10^4	/	4
Mott [3]	400	$>1.05 \times 10^5$	/

Mott [4]	263~3,846	/	/
Mott [5]	/	5×10^4	3
Filamentary [6]	1	10^6	3
This work	455	$>5 \times 10^6$	3

The lightweight design of our hardware stochastic *Roberts cross* operator not only downsizes the hardware scalability but also reduces the computational cost. We present in Fig. 6 the performance comparison between our and the conventional *Roberts cross* operator. The comparison is conducted in the same computational precision, i.e. each n -bit binary number is represented by 2^n -bit stochastic numbers. As shown in Fig. 6d, the stochastic operator requires fewer clock cycles to complete the edge detection with the stochastic number inputs <16 bits. As the bit length increases beyond 16 bits, our stochastic operator requires exponentially increased the clock cycles. This surpasses the conventional counterpart. Nevertheless, as presented in Fig. 6e, the stochastic operator maintains a lower energy consumption within 2,048 bits. To provide a more intuitive representation of the energy efficiency, we plot the energy consumption ratio of our stochastic operator to the conventional counterpart in Fig. 6f. This proves that **with 4-bit stochastic number inputs, our stochastic operator can consume ~95% less energy**. Similarly, with 64-bit stochastic number inputs, it can still consume ~90% less energy.

Figure 6. Performance and energy consumption comparison between the stochastic and conventional Roberts cross operators. Circuit designs of (a) our stochastic and (b) the conventional Roberts cross operator. The 2^n -bit inputs to our stochastic Roberts operator *are* stochastic numbers. (c) Circuit design of a n -bit subtractor and adder in (b). Relation of required (d) clock cycles and (e) energy consumption with respect to the bit length of the binary numbers and stochastic numbers. (f) Energy consumption ratio in (e). When

the length of the stochastic numbers is within 2,048 bits, the energy consumption of the stochastic Roberts operator is lower. All comparisons are conducted at a frequency of 100 kHz for demonstration purposes.

1.3) Scalability constrained by fabrication and testing conditions

While our proof-of-concept stochastic computing proves excellent performance in edge detection (as presented in our revised manuscript), the demonstration has been limited in the scalability. The scalability limitation comes down to the device fabrication, circuit integration, and parallel signal processing in the lab-based research.

In our lab-based research, we fabricate the memristors from solution-processed hBN via photolithography (Fig. S1-2). Though sampling test prove a high success rate and a uniform yet stochastic switching behavior in our fabrication (Fig. S3), large-scale manufacturing of memristors with a high-level uniformity is challenging, given the limited resources in lab-based research. Towards real-world memristor-based computing applications, we do believe the small-scale fabrication in the current memristor research and technologies needs to be substantially optimised (e.g. *Aguirre et al., Nature Communications 15.1 (2024): 1974*; *Huang et al., Nature Reviews Electrical Engineering (2024): 1-14*). This requires efforts from both the academia and industry to advances the underlying material sciences and the device technology as well as the manufacturing.

Figure S3. Device-to-device stochasticity test. (a) 12×12 memristor array in a crossbar configuration, with a fabrication yield approximating 100% (replotted from Fig. 2b), and (b) the corresponding schematic array showing the devices randomly selected for the sampling test. (c) Distributions of the measured V_{hold} and V_{th} of sampled devices for 20 sweeping cycles, along with the corresponding Gaussian fittings, indicating minimal device-to-device stochasticity – V_{hold} variation 6.6%, V_{th} variation 7.4%. The device-to-device variations are defined using the standard deviations of the mean V_{hold} (V_{th}) values.

The small scale memristor fabrication in the lab-based research further limits the scalability in integration of the memristors with the logic circuits towards the implementation of large scale edge detection operators, although the design

of the edge detection operators is highly compact and lightweight and, importantly, readily scaled up. We believe that, when large-scale manufacturing of memristors is realized, large-scale design and implementation of stochastic logic circuits and computing hardware can be realized.

Besides the above limitation in the fabrication scalability of memristors in lab-scaled research, performing large-scale parallel computing poses challenges. To perform large-scale parallel computing, large-scale parallel signal generation and probing systems are required. However, this can be quite costly and poses challenges to lab-based research. In our lab-based research, we have a signal generation and probing system that allow parallel generation and processing of 4 channels of signals. Therefore, we design and implement a hardware *Roberts cross* operator with 4 channels of parallel inputs (Fig. S11).

Given the above constrains, we perform small scale stochastic edge detection and computing in our work, as proof-of-concept exploration. Again, the feasibility in implementing large-scale stochastic computing would have to rely on simulation (Supplementary Video 1-3).

To clarify and address the possible queries from the readers on the scalability, we have now included the following discussion in the revised manuscript:

(Line 305) Though promising, large-scale stochastic edge detection requires efforts for realising large-scale design and fabrication of the memristors and peripheral electronic circuits, integration of the memristors with peripheral circuits, and parallel operation of the large-scale circuits. Amongst this, the success rate and uniformity of the memristors are still key concerns in large-scale manufacturing in current technological advances. A device-to-device non-uniformity can significantly impact the overall operation and performance of the circuits. A system-level analysis of memristors, SNEs, and peripheral circuits may therefore be adopted, e.g. the *Process-Voltage-Temperature* analysis. Hardware and algorithm codesigns are also needed to address or accommodate the non-idealities, e.g. noises and delays from the memristors and electronic circuits.

(Line 358) Limited by the scalability of the lab-based realization of the stochastic Roberts cross operator and parallel signal generation and testing, large-scale edge detection is conducted via simulation in Python3.

2) The memristor role in stochastic computing

Our memristors play a critical role in the hardware implementation of stochastic computing, given that 1) their fast switching, low switching energy, high endurance, and intrinsic stochasticity facilitate the efficient stochastic number encoding and processing (Fig. 2 and 3), 2) offer robust error-tolerance against bit-flips in the edge detection task (Fig. 5), and also 3) they reduce the computational cost and energy consumption (Fig. 6).

2.1) Efficient and stable stochastic number encoding and processing enabled by using the memristors

Our memristor-based stochastic number encoders (SNE) have demonstrated performance surpassing the other studies in terms of the encoding speed (~ 455 kbit/s), the encoding energy (~ 33 fJ per bit), the endurance ($>5 \times 10^6$ cycles), and the counts of required electrical components (3 transistor/memristor count) (Table S1). Particularly, we point out that **our memristor-based SNEs require three orders of magnitude fewer electrical components** compared to those conventionally realized with electronic circuits (e.g. those based on linear feedback shift registers), and that **our memristor-based SNEs demonstrate an encoding speed and endurance outperforming other memristors-based encoder reports**. Besides, the *Ornstein-Uhlenbeck* stochastic process modelling in Fig. S5 and the high switching endurance in Fig. S6 guarantee the efficient and stable stochastic number encoding and processing in the long run from our memristors.

Table S1. Comparison between the stochastic number encoders implemented using linear feedback shift registers (LFSR), magnetic tunnel junction (MTJ), Mott memristors, filamentary memristors, and our memristors. Data in the Table is reported in [1] *Borders et al., Nature 573.7774 (2019): 390-393*; [2] *Knag et al., IEEE Transactions on Nanotechnology 13.2 (2014): 283-293*; [3] *Seo et al., Advanced Materials (2024): 2402490*; [4] *Rhee et al., Nature Communications 14.1 (2023): 7199*; [5] *Deng et al., IEEE Electron Device Letters 44.10 (2023): 1776 – 1779*; [6] *Woo et al., Nature Communications 13.1 (2022): 5762*.

	Max encoding speed (kbit/s)	Endurance (cycles)	Transistor/memristor count
LFSR [1]	10^7	/	1194
LFSR [2]	10^5	/	/
MTJ [1]	10^4	/	4
Mott [3]	400	$>1.05 \times 10^5$	/
Mott [4]	263~3,846	/	/
Mott [5]	/	5×10^4	3
Filamentary [6]	1	10^6	3
This work	455	$>5 \times 10^6$	3

Figure S5. Stability test of the memristor switching stochasticity. Ornstein-Uhlenbeck process modeling on the measured threshold voltage V_{th} of the hBN filamentary memristor across the 1,000 consecutive sweeping cycles present in Fig. 2d. The experimental V_{th} data points well fit those from the Ornstein-Uhlenbeck process modelling, where $dV_{th,t} = \theta(\mu - V_{th,t}) + \sigma dW_t = 0.306 \times (0.729 - V_{th,t}) + 0.284 \times dW_t$. An Ornstein-Uhlenbeck process describes a stochastic process in a dynamical system (2). A dW_t denotes the variation of a Wiener process, i.e. a real-valued continuous-time stochastic process. This proves the stability of the switching stochasticity of our memristors in prolonged switching operations. At around 900-th cycle, the device is momentarily stuck at the high resistance states and thus hard to switch on probably due to the ambient disturbances but then returns to normal.

Figure S6. Endurance test. Endurance test of a typical memristor undergoing 5×10^6 consecutive test cycles under pulsed stimuli. For each test cycle, a $20 \mu\text{s}$ voltage pulse of 10 V is set to fully program the memristor and an $80 \mu\text{s}$ voltage pulse of 0.1 V is set to read the output. The output of the memristor is amplified by an operational amplifier and measured by an oscilloscope. The high (i.e. off) and low (i.e. on) resistance states in each test cycle are computed based on the oscilloscope measurement, and plotted in blue and red, respectively. Both the states remain stable throughout the full test.

2.2) Robust error-tolerance against bit-flips enabled by using the memristors

As we discuss in our revised manuscript, bit flips and other bit soft errors are prevalent in digital computing and neuromorphic computing, particularly, the memristor-based computing solutions considering the inherent stochasticity in memristor switching. Advanced error detection and correction techniques, such as parity bit, cyclic redundancy check, and hash function, are therefore required and widely adopted to address these errors. However, this inevitably incurs excessive hardware and computational cost. Other than substantially optimising the current memristor technologies, **adopting stochastic computing by leveraging the intrinsic stochasticity shows promise to accommodate the errors towards real-world computing applications.**

Our memristor-based stochastic computing solution exhibits excellent error-tolerance capacity against bit-flips (Fig. 5 and Fig. S12). Evaluated by the SSIM and PSNR metrics, our hardware stochastic *Roberts cross* operator demonstrates successful edge detection (SSIM >0.95 and PSNR >30 dB) in varying bit-flip injection cases (bit-flip ratio ranging from 5% to 50%). In comparison, the performance from the standard algorithmic method substantially degrades at a bit-flip injection of only 5%, with the edges hardly recognized and the SSIM and PSNR significantly decreased. Therefore, we conclude that **our memristor-based stochastic computing can naturally handle bit soft errors, as a particular advantage to computing enabled by the memristors**. This robust error-tolerance capability against bit-flips is well-suited for deployment in edge scenarios where information and input can be easily corrupted by the environmental noises and errors.

Figure 5. Error-tolerance test. (a) Error-tolerance test of the stochastic cross operator, showing bit-flips are injected into the original stochastic numbers for the tests. (b) Stochastic and (c) standard edge detection results and the corresponding SSIM maps of the first frame with bit-flip injection at a ratio of 0%, 5%, 10%, 20%, and 50%. For the stochastic edge detection, the high SSIM (>90%) and PSNR (>30 dB) prove that the bit-flip injection does not degrade the edge detection performance. In contrast, a low level of bit-flip injection significantly degrades the performance of the standard algorithmic edge detection. See Fig. S12 for the SSIM map of the standard edge detection, and the error-tolerance results at more bit-flip injection levels. Performance comparison in the (d) SSIM and (e) PSNR between the stochastic and standard edge detection results.

Figure S12. Error-tolerance test. (a) Stochastic and (b) standard edge detection results and the corresponding SSIM maps with bit-flip injection at a ratio of 0%, 2.5%, 5%, 10%, 20%, 30%, 40%, and 50%. For the stochastic edge detection, the high SSIM (>90%) and PSNR (>30 dB) prove that the bit-flip injection does not degrade the edge detection performance. In contrast, a low level of bit-flip injection significantly degrades the performance of the standard algorithmic edge detection.

2.3) Significantly reduced computational cost and energy consumption in edge detection enabled by using the memristors

As discussed, our memristor-based edge computing is lightweight, with reduced computational cost and energy consumption. We present in Fig. 6 the performance comparison between our and the conventional *Roberts cross* operator. The comparison is conducted in the same computational precision,

i.e. each n -bit binary number is represented by 2^n -bit stochastic numbers. As shown in Fig. 6d, the stochastic operator requires fewer clock cycles to complete the edge detection with the stochastic number inputs <16 bits. As the bit length increases beyond 16 bits, our stochastic operator requires exponentially increased the clock cycles. This surpasses the conventional counterpart. Nevertheless, as presented in Fig. 6e, the stochastic operator maintains a lower energy consumption within 2,048 bits. To provide a more intuitive representation of the energy efficiency, we plot the energy consumption ratio of our stochastic operator to the conventional counterpart in Fig. 6f. This proves that **with 4-bit stochastic number inputs, our stochastic operator can consume ~95% less energy**. Similarly, with 64-bit stochastic number inputs, it can still consume ~90% less energy.

Figure 6. Performance and energy consumption comparison between the stochastic and conventional Roberts cross operators. Circuit designs of (a) our stochastic and (b) the conventional Roberts cross operator. The 2^n -bit inputs to our stochastic Roberts operator *are* stochastic numbers. (c) Circuit design of a n -bit subtractor and adder in (b). Relation of required (d) clock cycles and (e) energy consumption with respect to the bit length of the binary numbers and stochastic numbers. (f) Energy consumption ratio in (e). When the length of the stochastic numbers is within 2,048 bits, the energy consumption of the stochastic Roberts operator is lower. All comparisons are conducted at a frequency of 100 kHz for demonstration purposes.

Regarding the comment of the reviewer – ‘*excessive reliance on additional circuit components*’, we need to point out that the memristors are one of the four types of fundamental electrical components, just like the resistors and capacitors (Strukov *et al.*, *Nature* (2008) 453, 80–83). This means that **the memristors cannot work alone in designing the circuits and completing**

the computational tasks. In our work, we are interested in exploiting the intrinsic stochasticity of memristors and using the stochasticity in realizing lightweight, error-tolerant stochastic computing. To achieve this, we need to integrate the memristors with other necessary peripheral components in the designs of stochastic number encoders, logic circuits, and operators. Indeed, research in memristor-based neuromorphic computing now focus on the integration of the memristors in CMOS circuits (e.g. *Jain et al., Nature Communications 16.1 (2025): 2719*; *Huang et al., Nature Communications 16.1 (2025): 101*).

Therefore, based on the above justifications, we believe the role of memristors in our stochastic computing solution is not weakened at all; in fact, **the memristors serve as the critical core in enabling the stochastic computing.**

3) **The novelty of edge detection**

Computer vision is a field based on extracting visual cues from images. Amongst the visual cues, gradient (or specifically, contour and texture) is the most intuitive one to help understand images and make initial decisions. **Edge detection, as a foundational method to extract the gradient, is widely used in computer vision,** ranging from autonomous driving, virtual and augmented reality, medical imaging diagnosis, to industrial automation, and beyond.

In this context, edge detection is widely employed as a fundamental image pre-processing technique to extract the key early visual cues, such as the shallow features of color, contour and texture, for efficient image understanding and initial decision-making. **As edge detection serves as the backbone for image processing tasks, edge detection has always been at the forefront of research,** with focus on improving the computing efficiency and robustness towards the noises and errors for practical applications (*Cetinkaya et al, Proceedings of the IEEE/CVF Conference on Computer Vision and Pattern Recognition. 2024*; *Zhou et al., Proceedings of the IEEE/CVF Conference on Computer Vision and Pattern Recognition. 2024*; *Yang et al., IEEE Transactions on Industrial Informatics. 2024*).

We understand that novelty is not merely defined as whether the research has been done before, but whether this research can solve practical problems. Given the fundamental importance of edge detection for image processing and the wide range of applications, **we consider our research on edge detection is novel, and we believe our research findings can lead to impacts to the practical edge detection applications, particularly at edge scenarios.** While edge detection may be foundational, constrained computational resources and potential environmental noises and errors can result in varying performance and hinder efficient user-scene interaction and decision-making, particularly when deployed in edge scenarios. As demonstrated, our work achieves lightweight, error-tolerant stochastic computing by the stochasticity of memristors.

Beyond edge detection, we do explore other stochastic computing applications with our memristors, for instance, Bayesian inference and multimodal fusion for timely reliable decision-making (<https://doi.org/10.48550/arXiv.2412.06838>). Here we present the key preliminary results in Fig. R1-3. Figure R1 shows the schematic of Bayesian decision-making using the memristors-based stochastic logics. Figure R2 and R3 illustrate the hardware implementation of Bayesian inference and fusion operators for route planning and obstacle detection in autonomous driving, respectively. More results are ready for discussion in our following work.

Figure R1. Bayesian decision-making. (a) Road scene parsing in self-driving, showing the car of investigation decides lane changing based on Bayes theorem when considering all relevant traffic conditions, e.g. the incoming cars, the pedestrian, and the ambient light and weather conditions. (b-d) Hardware implementation diagrams of Bayes theorem with memristors, where the memristors are used to enable probabilistic data encoding and logic operations and as such, render timely reliable Bayesian decision making based on probabilistic computational rules.

Figure R2. Hardware implementation of Bayesian inference. (a) Memristor-enabled Bayesian inference operator and its workflow to perform Bayesian inference for route planning. Prior knowledge with a prior probability $P(A)$ is updated as the new information with probability $P(B)$ becomes available, yielding a reliable decision with a posterior probability $P(A|B)$. Here $P(A)$ represents the initial belief of a vehicle (blue) trying to cut in lanes, and $P(B)$ represents the probability of the vehicle spotting another vehicle (red) on the target lane. The Bayesian inference operator is implemented with the memristor-enabled probabilistic AND and MUX logics. The probabilistic logics are used to conduct multiplication and weighted addition operations respectively for functioning as the numerator and denominator conforming to the Bayesian inference theorem. The division is achieved with a probabilistic MUX plus a D-Flip-Flop, following a classic divider design for stochastic numbers, CORDIV. (b) Hardware test of the Bayesian inference operator to perform route planning for self-driving vehicles. $P(A)$ and $P(B)$ are manually set to initialise the operator. Stochastic numbers at the key nodes are plotted accordingly. Bit 0s and 1s are marked in blue and red. The result $P(A|B) > P(A)$ means the red vehicle in (a) can cut in lanes with higher confidence. This enhanced decision reliability is attributed to the comprehensive evaluation of both the vehicle itself and the traffic situation. Pairwise (c) Pearson and (d) SC correlations between the stochastic numbers in (b).

Figure R3. Hardware implementation of Bayesian fusion. (a) Memristor-enabled Bayesian fusion operator and its workflow to perform Bayesian fusion. Pre-trained neural networks tailored for single modalities receive RGB and thermal information and output single-modal obstacle detection decisions, denoted as $P(y|x_1)$ and $P(y|x_2)$, where y represents the detected obstacle, and x_1 and x_2 represent the RGB and thermal information, respectively. The Bayesian fusion operator fuses the single-modal decisions into a reliable decision, denoted as $p(y|x_1, x_2)$, using the Bayesian fusion theorem. The Bayesian fusion operator is implemented with the memristor-enabled probabilistic AND and MUX logics. The probabilistic logics are used to conduct multiplication and weighted addition operations respectively for functioning as the numerator and denominator conforming to the Bayesian fusion theorem. The division is achieved with a probabilistic MUX plus a D-Flip-Flop, following a classic divider design for stochastic numbers, CORDIV. (b) Obstacle detection results before and after the Bayesian fusion of the RGB and thermal information under the different visibility conditions during daytime and nighttime. Before fusion, single-modal (RGB or thermal) networks typically lose certain target obstacles and make decisions with insufficient confidence. The Bayesian fusion operator effectively overcomes the single-modal shortages, addresses the target-missing and low-confidence issues, and achieves much more accurate and reliable results.

4) Contribution of hardware-based solution of edge detection

Regarding the comment of the reviewer – ‘*further software processing is still required to complete the edge detection task*’, again, we would like to point out that the memristors as one of the fundamental electrical components cannot work alone to complete the computational tasks and further software processing is required to present and visualize the results.

In our work, **our hardware stochastic circuits and operator fully perform the extraction task of the edges, while software is used for result presentation and**

visualization. To specify, **the stochastic number output by the operator directly represent the edge intensity** – a higher-level output represents a stronger edge, while a lower-level output represent the background. In this case, an extra simple comparator next to the stochastic number output is sufficient to distinguish whether there is an edge. This then requires the use of software to represent and visualize the edge detection results. Nevertheless, the stochastic numbers allow for seamless connection between the edge detection and downstream tasks (e.g. spiking neural networks, *Koo et al., IEEE Transactions on Circuits and Systems I: Regular Papers 67.8 (2020): 2546-2555*).

To clarify and address the possible queries from the readers towards the function of the hardware-based solution, we have now included the following discussion in the revised manuscript:

(Line 356) In the edge detection demonstrations, the hardware is used to fully perform the edge extraction, and the following software is used to present and visualize the results.

Reviewer #4:

After checking the rebuttal letter, it seems to me the previous questions have not been addressed in full. The authors has conducted more rigorous study on device endurance, which can exceed 5×10^6 cycles and outperforms the current studies. However, this is still insufficient to support practical applications, and the paper needs to provide an explanation on this point. The paper's assessment of latency and energy consumption is not sufficiently convincing. The evaluation uses data from components on the PCB, but the data obtained in this way does not reflect the situation of the actual chip. Because the clock period of digital circuits can be very fast (\sim GHz) in real chips, but the switching time of the memristor in the paper is only 50ns (20MHz), so the clock periods of the two should not be equivalent. In addition, V_{in} and V_{ref} are required to be adjusted according to the input, and other components (such as DACs) are needed to give the voltages, and this part of the overhead cannot be ignored. Also, the authors only provide a comparison with other memristor-enabled implementations, lacking a comparison with efficient digital implementations. Another point is it cannot be seen from Figure R1 that no additional device calibration or circuit design is required. For example, the V_{th} and V_{hold} of device 4 and device 5 differ by more than 0.2V.

We thank the reviewer for this important comment. We understand that the reviewer is concerned with 1) the switching endurance of our memristors, 2) the uniformity in the memristors and the resulted calibration and circuit redesign cost, 3) the operating frequency of the memristors and the mismatch with the digital circuits, 4) the comparisons with the other digital/memristor-based studies, and 5) the energy consumption estimation. We now address the concerns with the below point-by-point responses:

1) Switching endurance

The reviewer is concerned that the switching endurance as demonstrated from our memristors (over 5×10^6 cycles; shown below in Fig. S6) is not sufficient.

Figure S6. Endurance test. Endurance test of a typical memristor undergoing 5×10^6 consecutive test cycles under pulsed stimuli. For each test cycle, a 20 μ s voltage pulse of 10 V is set to fully program the memristor and an 80 μ s voltage pulse of 0.1 V is set to read the output. The output of the memristor is amplified by an operational amplifier and measured by an oscilloscope. The high (i.e. off) and

low (i.e. on) resistance states in each test cycle are computed based on the oscilloscope measurement, and plotted in blue and red, respectively. Both the states remain stable throughout the full test.

In our work, the stochastic number encoding is implemented in two stages – 1) the inputs are first encoded into pulsed signals of varying intensity, and 2) the pulsed signals are then processed into stochastic numbers by the memristors (Fig. 2). In our edge detection application, as demonstrated in Fig. 4, a **4-bit stochastic number encoding strategy** can already allow successful stochastic edge detection, though with poor yet acceptable performance (as evidenced by the PSNR and SSIM metrics). This means **the individual pixel per image frame requests only four switches of the memristors** to perform the stochastic edge detection in our demonstrations. Increasing the bit length of the stochastic numbers can substantially improve the edge detection performance. A stochastic number encoding resolution of 100-bit can ensure ideal stochastic edge detection performance (Fig. 4). A 100-bit stochastic number encoding can allow an image processing speed exceeding 1,000 frames per second, far surpassing the requirements for the current practical edge detection applications. In this context, therefore, **a switching endurance of 5×10^6 cycles of our memristors can sufficiently allow our lab-scaled demonstration of stochastic computing.**

Figure 4. Stochastic edge detection. (a) The exemplified image, i.e. the first frame of the 'The Horse in Motion', for edge detection demonstration. The region as marked is used to illustrate the edge detection process with the operator. The pixels in 0-255 grayscale are encoded into 100-bits. (b) Schematic stochastic Roberts cross operator, consisting of two SNEs, two XORs, and one MUX. See Fig. S11 for the hardware realization of the operator. (c) Gradient map yielded from scanning

with the operator, showing successful edge detection. (d) Edge detection of the first frame with the operator, and (e) the corresponding structural similarity index measure (SSIM) maps and peak signal-to-noise ratios (PSNR). The pixels are encoded into 4, 16, 64, and 256-bits as the inputs. The SSIM and PSNR show that the operator using more bits gives higher edge detection precision. For comparison, the edge detection performed using the standard algorithmic method is presented as the ground truth.

Here we note that **the switching endurance of our memristors (over 5×10^6 cycles) outperforms state-of-the-art reports**, e.g. **10^6 cycles** in *Cheong et al., Nature Communications 15.1 (2024): 6318*, **10^6 cycles** in *Woo et al., Nature Communications 15.1 (2024): 3245*, and **600 cycles** in *Teja et al., Nature Communications 15.1 (2024): 2334*. We can further elongate the cycling test to present a longer switching endurance, but we believe an endurance of 5×10^6 cycles can already prove the switching endurance and reliability of our memristors.

However, indeed, as commented by the reviewer, towards real-world memristor-based computing applications, we do believe **the current memristor technologies and fabrication methods (including ours) need to be further substantially optimised to further enhance the endurance and decrease the noise**, etc (e.g. *Aguirre et al., Nature Communications 15.1 (2024): 1974*; *Huang et al., Nature Reviews Electrical Engineering (2024): 1-14*). This requires efforts from both the academia and industry to advance the underlying material sciences and the device technology as well as the manufacturing.

To clarify and address the possible queries from the readers towards the switching endurance of our memristors, we have now included the following discussion in the revised manuscript:

(Line 127) Indeed, the endurance test for over 5×10^6 cycles proves a highly stable yet stochastic switching of our memristors (Fig. S6), **outperforming state-of-the-art reports (23–25) and allowing for a reliable integration of our memristors into circuits for implementing stochastic computing.**

(Line 305) **Though promising, large-scale stochastic edge detection requires efforts for realising large-scale design and fabrication of the memristors and peripheral electronic circuits, integration of the memristors with peripheral circuits, and parallel operation of the large-scale circuits. Amongst this, the success rate and uniformity of the memristors are still key concerns in large-scale manufacturing in current technological advances. A device-to-device non-uniformity can significantly impact the overall operation and performance of the circuits. A system-level analysis of memristors, SNEs, and peripheral circuits may therefore be adopted, e.g. the *Process-Voltage-Temperature* analysis. Hardware and algorithm codesigns are also needed to address or accommodate the non-idealities, e.g. noises and delays from the memristors and electronic circuits.**

(Line 358) Limited by the scalability of the lab-based realization of the stochastic Roberts cross operator and parallel signal generation and testing, large-scale edge detection is conducted via simulation in Python3.

2) Device uniformity

The reviewer is concerned with the device uniformity and the resulted calibration and circuit redesign when implementing the stochastic logic circuits; and that the device calibration and circuit redesign can incur excessive overhead.

As shown in our sampling test (Fig. S3), our memristor fabrication yield is approaching 100%, and the device-to-device variations are only 6.6% in V_{hold} and 7.4% in V_{th} . The low device-to-device variations, plus the high fabrication yield, allow us to design and implement stochastic computing. Indeed, we build the SNEs and stochastic operators with two randomly selected memristors as fabricated, and we can thus perform successful stochastic computing for edge detection. **No specific or additional device calibration or circuit redesign are required in our lab-scaled SNE and stochastic operator implementation.**

Figure S3. Device-to-device stochasticity test. (a) 12×12 memristor array in a crossbar configuration, with a fabrication yield approximating 100% (replotted from Fig. 2b), and (b) the corresponding schematic array showing the devices randomly selected for the sampling test. (c) Distributions of the measured V_{hold} and V_{th} of sampled devices for 20 sweeping cycles, along with the corresponding Gaussian fittings, indicating minimal device-to-device stochasticity – V_{hold} variation 6.6%, V_{th} variation 7.4%. The device-to-device variations are defined using the standard deviations of the mean V_{hold} (V_{th}) values.

Here we note that **the device-to-device variations in our memristors are minimal as compared to state-of-the-art reports of memristor fabrications** (e.g. $\sim 6\%$ in Jeong *et al.*, *Nature Electronics* (2025): 1-11; $\sim 6\%$ in Jeon *et al.*, *Nature communications* 15.1 (2024): 129).

However, again, we do need to point out that **the current memristor technologies need to be substantially optimised towards real-world neuromorphic computing applications.** To further enhance the uniformity towards practical real-

world circuit designs, it demands collaborative efforts from both the academia and industry to advance the underlying material sciences and the device technology (e.g. *Aguirre et al., Nature Communications 15.1 (2024): 1974*; *Huang et al., Nature Reviews Electrical Engineering (2024): 1-14*). Advances from the manufacturing (by e.g. *Process-Voltage-Temperature* analysis) and hardware-algorithm codesigns are also requested to accommodate the variations and errors in computing.

To clarify and address the possible queries from the readers towards the device calibration and circuit redesign, we have now included the following discussion in the revised manuscript:

(Line 352) Given the high device uniformity and fabrication yield, the operator is built with two randomly selected memristor based SNEs. The two SNEs exhibit similar $P_{\text{uncorrelated}}-V_{\text{in}}$ relation. No specific or additional memristor calibration, circuit redesigns, or testing of the $P_{\text{uncorrelated}}-V_{\text{in}}$ relations are required to implement the SNEs or the stochastic operators.

(Line 305) Though promising, large-scale stochastic edge detection requires efforts for realising large-scale design and fabrication of the memristors and peripheral electronic circuits, integration of the memristors with peripheral circuits, and parallel operation of the large-scale circuits. Amongst this, the success rate and uniformity of the memristors are still key concerns in large-scale manufacturing in current technological advances. A device-to-device non-uniformity can significantly impact the overall operation and performance of the circuits. A system-level analysis of memristors, SNEs, and peripheral circuits may therefore be adopted, e.g. the *Process-Voltage-Temperature* analysis. Hardware and algorithm codesigns are also needed to address or accommodate the non-idealities, e.g. noises and delays from the memristors and electronic circuits.

(Line 358) Limited by the scalability of the lab-based realization of the stochastic Roberts cross operator and parallel signal generation and testing, large-scale edge detection is conducted via simulation in Python3.

3) Operating frequency

The reviewer is concerned with the mismatch of our memristor switching frequency (up to 20 MHz) and the clock frequency of digital circuits (~GHz) when operating our memristors. Our discussion on the memristor-based stochastic computing strategy might not be thorough enough in the revised manuscript, and this might have led to misunderstanding of the reviewer on our stochastic number encoding and computing strategy.

In our memristor-based stochastic computing strategy, the stochastic number encoding is implemented in two stages – 1) the inputs are first encoded into pulsed signals of varying intensity, and 2) the pulsed signals are then processed into stochastic numbers by the memristors (Fig. 2). **The pulsed signals are configured as 100 kHz, far below the switching of the memristors and the clock frequency**

of the digital clocks. The clock frequency of digital circuits (~GHz) is not applied to the stochastic number encoding process.

We note that the switching of the memristors (20 MHz) does set an upper limit of stochastic number encoding speed. However, **a 20 MHz frequency is far beyond the stochastic number encoding frequency as required for practical edge detection applications.** For example, a 100-bit stochastic number encoding speed with the input bit stream encoding frequency (100 kHz) can in principle easily achieve edge detection with a frame rate of 1,000 frames per second. This well fulfils the requirements of practical edge detection applications ranging from autonomous driving and virtual and augmented reality to industrial automation and medical imaging diagnosis.

To clarify and address the possible queries from the readers towards the operating frequency, we have now made the following revisions in the revised manuscript:

(Line 129) We integrate the memristors into the circuits to develop the SNEs (Fig. 2a). **When in operation, signals in both digit and analog forms are first encoded into pulsed inputs, V_{in} , and then processed into stochastic numbers via the SNEs, as regulated by V_{ref} .**

(Line 146) **Here we note the encoding frequency of V_{in} is typically configured as 100 kHz, far below the switching of the memristors (up to 50 ns, or equivalently 20 MHz) and the clock frequency of the digital circuits (~GHz). This ensures that the SNEs can be applied in implementation of stochastic computing hardware and applications.**

(Line 520) **... All comparisons are conducted at an input encoding frequency of 100 kHz.**

4) Comparisons with the other studies

The reviewer is concerned with the lack of comparison of our work with the other studies.

In fact, as shown in Table S1, we have compared our memristor-enabled SNEs with 1) efficient digital implementations, e.g. based on *Linear Feedback Shift Registers* (LSFR), and 2) other memristor-based SNEs, in terms of the maximum encoding speed, endurance, and transistor/memristor count.

Table S1. Comparison between the stochastic number encoders implemented using linear feedback shift registers (LFSR), magnetic tunnel junction (MTJ), Mott memristors, filamentary memristors, and our memristors. Data in the Table is reported in [1] Borders et al., Nature 573.7774 (2019): 390-393; [2] Knag et al., IEEE Transactions on Nanotechnology 13.2 (2014): 283-293; [3] Seo et al., Advanced Materials (2024): 2402490; [4] Rhee et al., Nature Communications

14.1 (2023): 7199; [5] Deng et al., *IEEE Electron Device Letters* 44.10 (2023): 1776–1779; [6] Woo et al., *Nature Communications* 13.1 (2022): 5762.

	Max encoding speed (kbit/s)	Endurance (cycles)	Transistor/memristor count
LFSR [1]	10^7	/	1194
LFSR [2]	10^5	/	/
MTJ [1]	10^4	/	4
Mott [3]	400	$>1.05 \times 10^5$	/
Mott [4]	263~3,846	/	/
Mott [5]	/	5×10^4	3
Filamentary [6]	1	10^6	3
This work	455	$>5 \times 10^6$	3

5) Energy consumption estimation

The reviewer is concerned that the energy consumption estimation may not be accurate, and the energy overhead from the circuit operations with V_{in} and V_{ref} may have been missed.

Here we clarify that we have already considered the overhead from V_{in} and V_{ref} operations in the energy consumption estimation:

- V_{in} : When we estimate the switching energy consumption of the memristors, we adopt a worst-case estimation strategy. We assume $V_{in}=2V$ (consistent with Fig. S4), although in fact, a V_{in} of 1.1-1.5V can already perform the stochastic number encoding successfully (as demonstrated in Fig. 2).

Figure S4. Switching speed and energy consumption. (a) Electrical response to 1 μs pulsed signal, showing (b) a switching time of ~50 ns, a switching energy consumption of ~33 fJ, and a relaxation time of ~1,200 ns.

Figure 2. Stochastic number encoder (SNE). ... (f) $P_{\text{uncorrelated}} - V_{\text{in}}$ relation of a typical SNE in uncorrelation, fitting a sigmoid function $P_{\text{uncorrelated}} = 1/(1 + \exp[-38.9(V_{\text{in}} - 1.34)])$. (g) $P_{\text{positive}} - V_{\text{in}}$ and $P_{\text{negative}} - V_{\text{in}}$ relations of the SNE in positive and negative correlations. $P_{\text{negative}} - V_{\text{in}}$ fits a sigmoid function $P_{\text{negative}} = 1/(1 + \exp[-63.1(V_{\text{in}} - 0.19)])$ and $P_{\text{positive}} = 1 - P_{\text{negative}}$.

- V_{ref} : The overhead of V_{ref} is already included (and actually integral part of) in the overall power consumption of the comparator modules. Given this, we discuss the overall power consumption of the comparator modules (as presented in Table S3), instead of specifically estimating the energy consumption from V_{ref} operations.

Table S3. Power consumption of logic gate chips. The data are from the product datasheets.

	Chip	Power (mW)	Channel	Power/channel (mW)
AND	SN74HC08N	120.0	4	30.0
OR	HD74LS32P	25.7	4	6.4
XOR	HD74LS86P	32.0	4	8.0
MUX	SN74LS157N	96.0	4	24.0
Comparator	LM393	14.4	2	7.2

To clarify and address the possible queries from the readers towards the energy overhead in the energy consumption estimation, we have now made the following revisions in the revised manuscript:

(Line 255) In terms of the memristors, the switching energy is estimated as ~ 33 fJ per bit (Fig. S4). As such, encoding a 2^n -bit stochastic number consumes $\sim 33 \times 2^n$ fJ. Specifically, this estimation assumes the worst-case scenario, where a sufficiently large V_{in} of 2V is adopted. In fact, a V_{in} of 1.1-1.5V is adequate to perform the stochastic number encoding (Fig. 2f). In terms of the remaining circuits, here we estimate the energy consumption based on the required counts of logic gates and clock cycles for the stochastic edge detection operators, following $W = k \times T_c \times P$, where T_c is the clock cycle, k is the required counts of T_c , and P is the total power of the remaining electrical components, including the comparators and logic gates.

Further comments made by the reviewer as included in reviewer's review attachment:

This paper reports an energy-efficient scheme for edge detection using memristor-enabled stochastic logics, while being highly tolerant of bit flipping. The stochastic logics are implemented by integrating logic gates and stochastic number encoders (SNE) using memristors and are used to achieve a highly compact Roberts cross operator. However, the paper may not be suitable for publication due to the following concerns.

1. One of the major strengths highlighted in this paper is the high tolerance for bit-flips, but the comparison in the paper is only done with conventional digital circuits, which is not reasonable. Conventional digital circuits hardly ever suffer from bit-flipping, and the problem is only noticeable in memristor-based computing circuits.

We thank the reviewer for this comment.

We compare our work with digital circuits in terms of tolerance for bit-flips based on two major considerations: 1) there are very few studies yet thus far on hardware implementation of stochastic computing with memristors and as such, we fail to provide a detailed comparison of our work with the other memristor-based studies in terms of tolerance for bit-flips; and 2) there are digital circuit implementations of stochastic computations and, importantly, the digital circuits and computing can in general serve as the benchmarks for memristor-based stochastic computing studies. Therefore, **we provide the comparison of our work with the digital circuits**, and we believe **the comparison can reasonably reflect the advances of our memristor-based stochastic computing in error-tolerant capacity**.

Regarding whether bit flips are still common in digital circuits and computing, we would argue that **bit flips are still a major concern in digital circuits and computing** (e.g. *Miskov-Zivanov et al., IEEE Transactions on Computer-Aided Design of Integrated Circuits and Systems 29.10 (2010): 1614-1627*; *Mukherjee et al., 11th International Symposium on High-Performance Computer Architecture. IEEE, 2005*). Bit flips can easily arise from noise and interference. **To address the bit flip problems, current digital circuits and computing commonly adopt advanced error detection and correction techniques**, e.g. parity bit, cyclic redundancy check, and hash functions. The use of the

advanced error detection and correction techniques can cancel the bit-flips and make the bit-flips seem not significant any more in digital circuits and computing. However, these error detection and correction techniques can inevitably incur excessive hardware and computational cost.

We note that bit flips and other bit soft errors are prevalent in memristor-based computing. Though stochastic computing shows promise to accommodate the errors, towards real-world neuromorphic computing applications in real-world, we again do point out that the current memristor technologies and fabrications need to be substantially optimised (e.g. Aguirre et al., Nature Communications 15.1 (2024): 1974; Huang et al., Nature Reviews Electrical Engineering (2024): 1-14). This demands collaborative efforts from both the academia and industry to advance the underlying material sciences and the device technology. Advances from the manufacturing (by e.g. *Process-Voltage-Temperature* analysis) and hardware-algorithm codesigns are also requested to accommodate the variations and errors in computing.

To clarify and address the possible queries from the readers towards the bit flips, we have now made the following revisions in the revised manuscript:

(Line 49) Bit-flips, as common soft errors in digital circuits and computing (8, 9), can be typically induced by noise and interferences (10). Though advanced error detection and correction techniques, such as parity bit, cyclic redundancy check, and hash function, are now widely adopted to address bit-flips, they inevitably incur excessive hardware and computational cost.

2. The paper's assessment of latency and energy consumption is not sufficiently convincing. The evaluation uses data from components on the PCB, but the data obtained in this way does not reflect the situation of the actual chip. Because the clock period of digital circuits can be very fast (~GHz) in real chips, but the switching time of the memristor in the paper is only 50ns (20MHz), so the clock periods of the two should not be equivalent. In addition, V_{in} and V_{ref} are required to be adjusted according to the input, and other components (such as DACs) are needed to give the voltages, and this part of the overhead should not be ignored.

We appreciate the reviewer for this comment. Our discussion on the memristor-based stochastic computing strategy might not be thorough enough in the revised manuscript, and this might have led to misunderstanding of the reviewer on our stochastic number encoding and computing.

In our memristor-based stochastic computing strategy, the stochastic number encoding is implemented in two stages – 1) the inputs are first encoded into pulsed signals of varying intensity, and 2) the bit streams are then processed into stochastic numbers by the memristors (Fig. 2). **The input bit stream encoding frequency is configured as 100 kHz, far below the switching of the memristors and the clock frequency of the digital clocks.** The clock frequency of digital circuits (~GHz) is not applied to our stochastic number encoding process.

Note that the switching of the memristors (20 MHz) does set an upper limit of stochastic number encoding speed. However, **a 20 MHz frequency is far beyond the stochastic number encoding frequency as required for practical edge detection applications.** For example, a 100-bit stochastic number encoding with the input bit stream encoding frequency (100 kHz) can in principle easily achieve edge detection with a frame rate of 1,000 frames per second. This well fulfils the requirements of practical edge detection applications ranging from autonomous driving and virtual and augmented reality to industrial automation and medical imaging diagnosis.

Regarding the energy overhead from the circuit operations with V_{in} and V_{ref} , **we clarify that we have already considered the overhead from V_{in} and V_{ref} operations in the energy consumption:**

- V_{in} : When we estimate the switching energy consumption of the memristors, we adopt a worst-case estimation strategy. We assume $V_{in}=2V$ (consistent with Fig. S4), although in fact, a V_{in} of 1.1-1.5V can already perform the stochastic number encoding successfully (as demonstrated in Fig. 2).
- V_{ref} : The overhead of V_{ref} is already included (and actually integral part of) in the overall power consumption of the comparator modules. Given this, we discuss the overall power consumption of the comparator modules (as presented in Table S3), instead of specifically estimating the energy consumption from V_{ref} operations.

To clarify and address the possible queries from the readers towards the clock frequency and hardware overhead, we have now made the following revisions in the revised manuscript:

(Line 129) We integrate the memristors into the circuits to develop the SNEs (Fig. 2a). **When in operation, signals in both digit and analog forms are first encoded into pulsed inputs, V_{in} , and then processed into stochastic numbers via the SNEs, as regulated by V_{ref} .**

(Line 146) Here we note the encoding frequency of V_{in} is typically configured as 100 kHz, far below the switching of the memristors (up to 50 ns, or equivalently 20 MHz) and the clock frequency of the digital circuits (~GHz). This ensures that the SNEs can be applied in implementation of stochastic computing hardware and applications.

(Line 255) In terms of the memristors, the switching energy is estimated as ~33 fJ per bit (Fig. S4). As such, encoding a 2^n -bit stochastic number consumes $\sim 33 \times 2^n$ fJ. **Specifically, this estimation assumes the worst-case scenario, where a sufficiently large V_{in} of 2V is adopted. In fact, a V_{in} of 1.1-1.5V is adequate to perform the stochastic number encoding (Fig. 2f).** In terms of the remaining circuits, here we estimate the energy consumption based on the required counts of logic gates and clock cycles for the stochastic edge detection operators, following $W =$

$k \times T_c \times P$, where T_c is the clock cycle, k is the required counts of T_c , and P is the total power of the remaining electrical components, including the comparators and logic gates.

(Line 520) ... All comparisons are conducted at an input encoding frequency of 100 kHz.

3. Device consistency is critical when building an SNE, otherwise it will result in the same input being regulated differently for different SNE, which requires additional overhead to store the relationship between V_{in}/V_{ref} and P for each SNE. Figure. S3 shows the results for device-to device data. From the results, it is clear that each SNE should have a different V-P correspondence, which will cost a lot.

We appreciate the reviewer for this important comment.

Regarding the device consistency, device consistency is indeed critical for the SNE and operator design. As shown in our sampling test (Fig. S3), our memristor fabrication yield is approaching 100%, and the device-to-device variations are only 6.6% in V_{hold} and 7.4% in V_{th} . The low device-to-device variations, plus the high fabrication yield, allow us to design and implement stochastic computing. Indeed, we build the SNEs and stochastic operators with two randomly selected memristors as fabricated, and we can thus perform successful stochastic computing for edge detection. **No specific or additional device calibration or circuit redesign are required in our lab-scaled implementation of SNEs and stochastic operators.**

Here we note that **the device-to-device variations in our memristors are minimal as compared to state-of-the-art advances of memristor fabrications** (e.g. *~6% in Jeong et al., Nature Electronics (2025): 1-11*; *~6% in Jeon et al., Nature communications 15.1 (2024): 129*). However, we again do point out that the current memristor technologies and fabrications need to be substantially optimised towards real-world neuromorphic computing applications (e.g. *Aguirre et al., Nature Communications 15.1 (2024): 1974*; *Huang et al., Nature Reviews Electrical Engineering (2024): 1-14*). To further enhance the uniformity towards practical circuit designs, it demands collaborative efforts from both the academia and industry to advance the underlying material sciences and the device technology. Advances from the manufacturing (by e.g. *Process-Voltage-Temperature analysis*) and hardware-algorithm codesigns are also requested to accommodate the variations and errors in computing.

To clarify and address the possible queries from the readers towards the device uniformity, we have now included the following discussion in the revised manuscript:

(Line 305) **Though promising, large-scale stochastic edge detection requires efforts for realising large-scale design and fabrication of the memristors and peripheral electronic circuits, integration of the memristors with peripheral circuits, and parallel operation of the large-scale circuits. Amongst this, the success rate and**

uniformity of the memristors are still key concerns in large-scale manufacturing in current technological advances. A device-to-device non-uniformity can significantly impact the overall operation and performance of the circuits. A system-level analysis of memristors, SNEs, and peripheral circuits may therefore be adopted, e.g. the *Process-Voltage-Temperature* analysis. Hardware and algorithm codesigns are also needed to address or accommodate the non-idealities, e.g. noises and delays from the memristors and electronic circuits.

(Line 358) Limited by the scalability of the lab-based realization of the stochastic Roberts cross operator and parallel signal generation and testing, large-scale edge detection is conducted via simulation in Python3.

Regarding the V-P correspondence, given the low device-to-device variations and high fabrication yield of our memristors as well as the lightweight SNE and operator designs, the two SNEs in the operator exhibit similar V-P correspondence. The V-P correspondence is obtained from a test, akin to a factory test, along with the operator design. **No additional V-P correspondences need to be retested for the SNE and operator designs in our lab-scaled implementation and, therefore, the overhead of the V-P correspondence tests is not considered in our work.**

To clarify and address the possible queries from the readers regarding the V-P correspondences, we have now made the following revision in the revised manuscript:

(Line 352) Given the high device uniformity and fabrication yield, the operator is built with two randomly selected memristor based SNEs. The two SNEs exhibit similar $P_{\text{uncorrelated}}-V_{\text{in}}$ relation. No specific or additional memristor calibration, circuit redesigns, or testing of the $P_{\text{uncorrelated}}-V_{\text{in}}$ relations are required to implement the SNEs or the stochastic operators.

4. SNE requires repeated switching of the device, and as can be seen in Figure. 6 a 10-bit binary number requires a 1024-bit stochastic number to achieve the same accuracy, which means that a single calculation requires ~102 rewrites of the device (already taking into account the stochastic switching of the device). Figure. S6 demonstrates an endurance of more than 5×10^6 , but compared to the frequency of repeated switching required, such an endurance seems to not guarantee that the SNE could work properly for a long time.

We appreciate the reviewer for this important comment. Again, our discussion on the memristor-based stochastic computing strategy may not be thorough enough in the revised manuscript, and this might have led to misunderstanding of the reviewer on our stochastic number encoding and computing.

A 10-bit binary number does require a 1024-bit stochastic number to represent, but **we do not actually adopt 1024-bit stochastic number encoding in performing the stochastic computing.** In our work, the stochastic number encoding is implemented in two stages –

1) the inputs are first encoded into pulsed signals of varying intensity, and 2) the bit streams are then processed into stochastic numbers by the memristors (Fig. 2). In our edge detection application, as demonstrated in Fig. 4, a **4-bit stochastic number encoding strategy** can already allow successful stochastic edge detection, though with poor yet acceptable performance (as evidenced by the PSNR and SSIM metrics). This means **the individual pixel per image frame requests only four switches of the memristors** to perform the stochastic edge detection. Increasing the bit length of the stochastic numbers can substantially improve the edge detection performance. A stochastic number encoding resolution of <100-bit can ensure ideal stochastic edge detection performance (Fig. 4). A 100-bit stochastic number encoding can allow an image processing speed exceeding 1,000 frames per second, far surpassing the requirements for the current practical edge detection. In this context, therefore, **a switching endurance of 5×10^6 cycles of our memristors is sufficient to allow our lab-scaled demonstration of stochastic computing.**

Here we note that **the switching endurance as tested in our memristors outperforms the state-of-the-art advances**, e.g. **10^6 cycles** in *Cheong et al., Nature Communications 15.1 (2024): 6318*, **10^6 cycles** in *Woo et al., Nature Communications 15.1 (2024): 3245*, and **600 cycles** in *Teja et al., Nature Communications 15.1 (2024): 2334*. **We can for sure further elongate the cycling test to present a longer endurance, but we believe an endurance of 5×10^6 cycles can already prove the switching endurance and reliability of our memristors.**

Nevertheless, towards real-world neuromorphic computing applications of memristors, we believe the current memristor technologies and fabrication methods need to be further substantially optimised to further enhance the endurance and decrease the noise, *etc.* (e.g. *Aguirre et al., Nature Communications 15.1 (2024): 1974*; *Huang et al., Nature Reviews Electrical Engineering (2024): 1-14*). This requires efforts from both the academia and industry to advance the underlying material sciences and the device technology as well as the manufacturing.

To clarify and address the possible queries from the readers towards the endurance of the memristors, we have now included the following discussion in the revised manuscript:

(Line 126) Indeed, the endurance test for over 5×10^6 cycles proves a highly stable yet stochastic switching of our memristors (Fig. S6), **outperforming state-of-the-art reports (23–25) and allowing for a reliable integration of our memristors into circuits for implementing stochastic computing.**

(Line 305) **Though promising, large-scale stochastic edge detection requires efforts for realising large-scale design and fabrication of the memristors and peripheral electronic circuits, integration of the memristors with peripheral circuits, and parallel operation of the large-scale circuits. Amongst this, the success rate and uniformity of the memristors are still key concerns in large-scale manufacturing in current technological advances. A device-to-device non-uniformity can significantly impact the overall operation and performance of the circuits. A**

system-level analysis of memristors, SNEs, and peripheral circuits may therefore be adopted, e.g. the *Process-Voltage-Temperature* analysis. Hardware and algorithm codesigns are also needed to address or accommodate the non-idealities, e.g. noises and delays from the memristors and electronic circuits.

(Line 358) Limited by the scalability of the lab-based realization of the stochastic Roberts cross operator and parallel signal generation and testing, large-scale edge detection is conducted via simulation in Python3.

This paper reports an energy-efficient scheme for edge detection using memristor-enabled stochastic logics, while being highly tolerant of bit flipping. The stochastic logics are implemented by integrating logic gates and stochastic number encoders (SNE) using memristors and are used to achieve a highly compact Roberts cross operator. However, the paper may not be suitable for publication due to the following concerns.

1. One of the major strengths highlighted in this paper is the high tolerance for bit-flips, but the comparison in the paper is only done with conventional digital circuits, which is not reasonable. Conventional digital circuits hardly ever suffer from bit-flipping, and the problem is only noticeable in memristor-based computing circuits.
2. The paper's assessment of latency and energy consumption is not sufficiently convincing. The evaluation uses data from components on the PCB, but the data obtained in this way does not reflect the situation of the actual chip. Because the clock period of digital circuits can be very fast (\sim GHz) in real chips, but the switching time of the memristor in the paper is only 50ns (20MHz), so the clock periods of the two should not be equivalent. In addition, V_{in} and V_{ref} are required to be adjusted according to the input, and other components (such as DACs) are needed to give the voltages, and this part of the overhead should not be ignored.
3. Device consistency is critical when building an SNE, otherwise it will result in the same input being regulated differently for different SNE, which requires additional overhead to store the relationship between V_{in}/V_{ref} and P for each SNE. Figure. S3 shows the results for device-to-device data. From the results, it is clear that each SNE should have a different V-P correspondence, which will cost a lot.
4. SNE requires repeated switching of the device, and as can be seen in Figure. 6 a 10-bit binary number requires a 1024-bit stochastic number to achieve the same accuracy, which means that a single calculation requires $\sim 10^2$ rewrites of the device (already taking into account the stochastic switching of the device). Figure. S6 demonstrates an endurance of more than 5×10^6 , but compared to the frequency of repeated switching required, such an endurance seems to not guarantee that the SNE could work properly for a long time.